



# Investigating differences in DOAS retrieval codes using MAD-CAT campaign data

Enno Peters[1], Gaia Pinardi[2], André Seyler[1], Andreas Richter[1], Folkard Wittrock[1],
Tim Bösch[1], John P. Burrows[1], Michel Van Roozendael[2], François Hendrick[2],
Theano Drosoglou[3], Alkis F. Bais[3], Yugo Kanaya[4], Xiaoyi Zhao[5], Kimberly Strong[5],
Johannes Lampel[6, 10], Rainer Volkamer[7], Theodore Koenig[7], Ivan Ortega[7], Ankie Piters[8],
Olga Puentedura[9], Mónica Navarro[9], Laura Gómez[9], Margarita Yela González[9],
Julia Remmers[10], Yang Wang[10], Thomas Wagner[10], Shanshan Wang[11, 12],
Alfonso Saiz-Lopez[11], David García-Nieto[11], Carlos A. Cuevas[11], Nuria Benavent[11],
Richard Querel[13], Paul Johnston[13], Oleg Postylyakov[14], Alexander Borovski[14],
Alexander Elokhov[14], Ilya Bruchkouski[15], Cheng Liu[16, 17, 18], Qianqian Hong[18], Haoran Liu[16],
Claudia Rivera[19], Michel Grutter[19], Wolfgang Stremme[19], M. Fahim Khokhar[20], and Junaid
Khayyam[20]

[1]Institute of Environmental Physics (IUPB), University of Bremen, Germany
[2]Royal Belgian Institute for Space Aeronomy (BIRA-IASB), Brussels, Belgium
[3]Aristotle University of Thessaloniki, Greece
[4]Japan Agency for Marine-Earth Science and Technology (JAMSTEC), Japan
[5]Department of Physics, University of Toronto, Ontario, Canada
[6]Institute of Environmental Physics, University of Heidelberg, Germany
[7]University of Colorado, Boulder, USA
[8]Royal Netherlands Meteorological Institute (KNMI) De Bilt, The Netherlands
[9]National Institute for Aerospace technology, INTA, Madrid, Spain
[10]Max Planck Institute for Chemistry, Mainz, Germany
[11]Department of Atmospheric Chemistry and Climate, Institute of Physical Chemistry Rocasolano, CSIC,
Madrid, Spain
[12]Shanghai Key Laboratory of Atmospheric Particle Pollution and Prevention (LAP$^3$), Department of
Environmental Science & Engineering, Fudan University, Shanghai, China
[13]National Institute of Water and Atmospheric Research (NIWA), Lauder, New Zealand
[14]A. M. Obukhov Institute of Atmospheric Physics, Russian Academy of Sciences, Moscow, Russia
[15]Belarusian State University (BSU), Minsk, Belarus
[16]School of Earth and Space Sciences, University of Science and Technology of China, Hefei, 230026, China
[17]CAS Center for Excellence in Regional Atmospheric Environment, Xiamen, 361021, China
[18]Key Lab of Environmental Optics & Technology, Anhui Institute of Optics and Fine Mechanics, Chinese
Academy of Sciences, Hefei, 230031, China
[19]National Autonomous University of Mexico (UNAM), Mexico
[20]Institute of environmental sciences and engineering (IESE), National university of sciences and technology
(NUST) Islamabad, Pakistan

## Abstract

The Differential Optical Absorption Spectroscopy (DOAS) method is a well-known re-
mote sensing technique that is nowadays widely used for measurements of atmospheric trace
gases, creating the need for harmonization and characterization efforts. In this study, an
intercomparison exercise of DOAS retrieval codes from 17 international groups is presented
focusing on $NO_2$ slant columns. The study is based on data collected by one instrument
during the Multi-Axis DOAS Comparison campaign for Aerosols and Trace gases (MAD-
CAT) in Mainz, Germany, in summer 2013. As data from the same instrument is used by
all groups, the results are free of biases due to instrumental differences, which is in contrast
to previous intercomparison exercises.

While in general an excellent correlation of $NO_2$ slant columns between groups of $> 99.98\%$
(noon reference fits), and $> 99.2\%$ (sequential reference fits) for all elevation angles is found,





differences between individual retrievals are as large as 8% for NO$_2$ slant columns and 100% for RMS residuals.

Two kinds of disagreements were identified: (1) Absolute slant column differences were found to result predominantly from the choice of the reference spectrum. (2) Relative differences were found to originate from the numerical approach for solving the DOAS equation as well as the treatment of the slit function.

Differences in the implementations of the intensity offset correction lead to disagreements for retrievals close to sunrise (8-10% for NO$_2$, 80% for RMS residual). Apart from this, the largest effect of $\approx$8% difference in NO$_2$ was found to arise from the reference treatment, in particular for fits using a sequential reference. In terms of RMS fit residual, the reference treatment has only a minor impact. In contrast, the wavelength calibration as well as the intensity offset correction were found to have the largest impact (up to 80%) on RMS residual while having only a minor impact on retrieved NO$_2$ slant columns.

# 1 Introduction

In this study, the consistency of Differential Optical Absorption Spectroscopy (DOAS) retrievals of tropospheric nitrogen dioxide (NO$_2$) from ground-based scattered light observations is evaluated. NO$_2$ is released into the atmosphere predominantly in the form of NO as a result of combustion processes at high temperatures. Through reaction of NO with ozone (O$_3$), NO$_2$ is rapidly produced and (during the day) back-converted into NO by photolysis. Therefore, nitrogen oxides are often discussed in terms of NOx (NO + NO$_2$ = NOx). NO$_2$ is a key species in the formation of tropospheric ozone and a prominent pollutant in the troposphere, causing (together with aerosols) the typical brownish colour of polluted air. In addition, it is harmful for lung tissue and a powerful oxidant. In the lower troposphere, the lifetime of NO$_2$ is short (several hours) due to reaction with OH and photo-dissociation, and thus it is found mostly close to its sources, making it a good tracer of local pollution. Both anthropogenic sources such as burning of fossil fuel in industry, power generation, and traffic as well as biogenic sources including bush and forest fires contribute to the tropospheric NOx loading. In addition, NO$_2$ is released from soil microbial processes and lightning events (Lee et al., 1997). As a result, high NO$_2$ amounts are mostly observed above industrialized and urban areas, traffic routes and over bush fires.

Using its characteristic absorption bands in the UV and visible spectral range, NO$_2$ has been successfully measured for many years using the DOAS technique (Brewer, 1973; Noxon, 1975; Platt, 1994) both from the ground and from space (e.g. Richter et al., 2005; Beirle et al., 2011). In addition, airborne and car-based measurements (e.g. Schönhardt et al., 2015; Shaiganfar et al., 2011), have been performed to close the gap between ground-based observations providing continuous temporal, but poor spatial resolution and satellite measurements which offer global observations, but only up to one measurement per day above each location. Using ground-based multi-axis (MAX)-DOAS measurements at different elevation angles, a more accurate vertical column (VC) can be retrieved with information on the vertical distribution of NO$_2$ and other trace gases in the troposphere (e.g. Hönninger et al., 2004; Wittrock, 2006; Frieß et al., 2011; Wagner et al., 2011).

As result of the viewing geometry, ground-based MAX-DOAS observations are most sensitive to the lowest layers of the troposphere. Here they provide high sensitivity and low relative measurement errors. Averaging over longer integration times can further reduce statistical noise. Another advantage is that MAX-DOAS stations can be operated automatically and usually require little maintenance. MAX-DOAS measurements were therefore performed in remote regions for the investigation of background concentrations and in many locations for the validation of satellite observations (e.g. Takashima et al., 2012; Peters et al., 2012).

There currently exists a variety of different instruments and retrieval codes designed to perform and analyse MAX-DOAS measurements. While the basic approaches are similar, differences exist which can potentially lead to inconsistencies in measurements from different instruments and retrievals. As this is an important limitation for the use of MAX-DOAS measurements in a global observing system, the Network for the Detection of Atmospheric Composition Change NDACC (formerly known as NDSC) has organised several intercomparison campaigns aimed at comparing instruments, retrievals,





and uncertainties of a wide range of DOAS instruments. The first of these campaigns were focused entirely on zenith-sky observations for stratospheric absorbers (Hofmann et al., 1995; Vaughan et al., 1997; Roscoe et al., 1999) but later also included other viewing directions (Vandaele et al., 2005). The

Cabauw Intercomparison of Nitrogen Dioxide Measuring Instruments (CINDI) campaign in 2009 in Cabauw, the Netherlands was the first to also have a focus on MAX-DOAS observations of tropospheric species (Roscoe et al., 2010; Piters et al., 2012; Pinardi et al., 2013) and has recently been followed by the CINDI-2 campaign performed in 2016, also in Cabauw. However, these intercomparison exercises concentrated mostly on results originating from different retrieval codes and instruments,

and separation of instrument and retrieval effects was not easily possible. In some cases, synthetic spectra were used to intercompare retrieval algorithms, and while such tests can highlight differences between retrieval approaches, they give no insight into the way different retrieval codes deal with instrumental effects such as intensity offsets, resolution changes or spectral drifts.

The present study was performed in the framework of the European FP7 project QA4ECV (Qual-

ity Assurance for Essential Climate Variables) which aims at providing quality assurance for satellite derived ECVs such as $NO_2$, HCHO and CO by characterizing the uncertainty budgets through uncertainty analysis and error propagation but also by validation with external data sets. In this context, ground-based MAX-DOAS measurements can play an important role, and harmonisation of the retrieval approaches and quality assurance for the reference measurements are needed. The results of

the analysis performed in the QA4ECV project are expected to contribute to the harmonisation of data from MAX-DOAS instruments, in particular for the ongoing integration of such measurements in the NDACC network.

The work reported here overcomes limitations from previous studies by using real measurements originating from a single instrument, facilitating the study of the agreement between different retrieval

codes on real data without instrumental bias. The intercomparison was performed on spectra recorded by the University of Bremen (IUPB) instrument during the Multi-Axis DOAS Comparison campaign for Aerosols and Trace gases (MAD-CAT) carried out in Mainz, Germany, in summer 2013 (`http://joseba.mpch-mainz.mpg.de/mad_cat.htm`). Data was distributed to 17 international groups working on ground-based DOAS applications. Each group analysed these spectra using their own DOAS

retrieval code but prescribed fit settings for fitting window, cross-sections, polynomial and offset correction. Resulting slant columns from all groups were then compared to IUPB results chosen as the arbitrary reference, evaluating the level of agreement and systematic differences and investigating their algorithmic origins. With this set-up, nearly all sources of disagreement were removed, and only

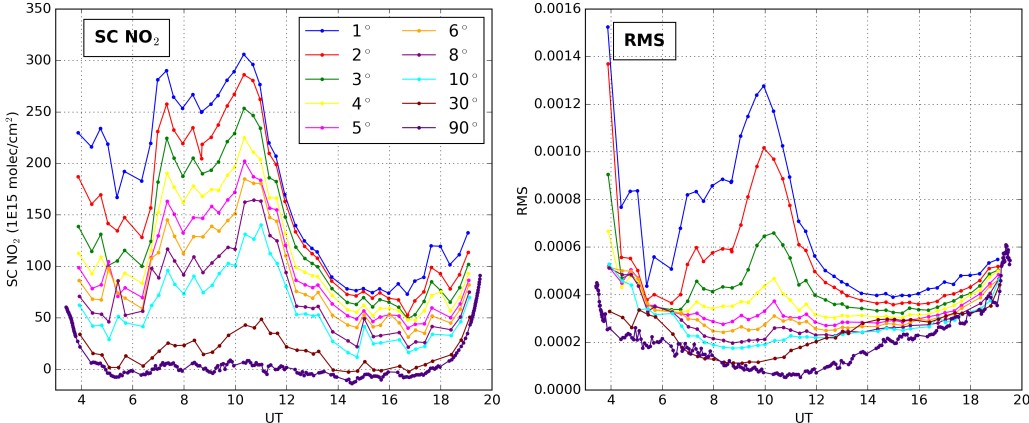

Figure 1: $NO_2$ slant columns and fit residual root mean square (RMS) obtained from the IUPB retrieval code for the intercomparison day used in this study. Different elevation angles are color-coded. The fit settings correspond to v1 settings as described in Sect. 2.4 (Tab. 1).



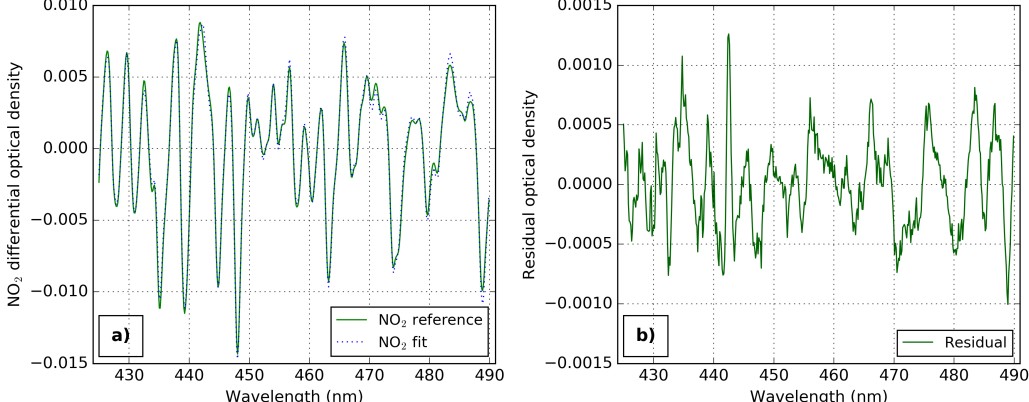

Figure 2: (a) Example of the fitted $NO_2$ optical density in 2° elevation angle at 14:40 UT corresponding to a slant column of $\approx$7E16 molec/cm$^2$ (compare Fig. 1). The solid line is the $NO_2$ optical density (differential cross section multiplied by the retrieved slant column) and the dashed line is the solid line plus the fit residual. The $NO_2$ optical density is much larger than the residual, which is explicitly shown in (b).

those differences between retrieval codes were investigated which are not usually reported.

This intercomparison exercise concentrates on one day (18 June 2013) during the MAD-CAT campaign having the best weather and viewing conditions. As an example, Fig. 1 shows $NO_2$ slant columns (left) and fit RMS residual (right) retrieved with the IUPB software. It is nicely seen that $NO_2$ slant columns measured at different elevation angles are separated as a result of differences in the light path. It should be mentioned that $NO_2$ slant columns are relatively large as a result of anthropogenic

pollution at the densely populated and industrialized measurement location and thus the findings of this study correspond to polluted urban environments. However, these are normally of interest for $NO_2$ observations. Fig. 1 also shows that the RMS of the fit residuals (in the following simply denoted as RMS) separates with elevation angles as well. In addition, the shape of the RMS in small elevation angles is very similar to that of $NO_2$ slant columns, indicating that predominantly $NO_2$ related effects

such as the wavelength dependence of the $NO_2$ AMF (which was not included in the fit shown here) limit the fit quality (note, this study aims not at finding best $NO_2$ fit settings but studying disagreements originating from different retrieval codes). This is supported by the observation that the shape of $NO_2$ slant columns is not seen in the RMS for larger elevation angles as effects related to the $NO_2$ absorption are smaller. In addition, the RMS is very large in the early morning for small elevations,

which is discussed in detail later and results from pointing towards sunrise. However, in general $NO_2$ is retrieved very accurately as Fig. 2 demonstrates, showing the $NO_2$ optical density and fit residual for a measurement in 2° elevation angle (this is a typical example fit).

The manuscript is structured in the following way: Sect. 2 provides details about the MAD-CAT campaign, the measurements, and the $NO_2$ intercomparison exercise as well as participating retrieval

codes. The comparison between results from the different groups is presented in Sect. 3. Sect. 4 attempts to reproduce observed differences between groups using the IUPB retrieval code with different settings in order to identify and quantify sources of disagreements. The manuscript ends with a summary and recommendations for better harmonisation of ground-based MAX-DOAS measurements.





## 2 Measurements

### 2.1 DOAS technique

The DOAS technique is based on Lambert-Beer's law which describes the attenuation of light passing through a medium. Here, it is applied to measurements of scattered sunlight. The spectral attenuation caused by scattering is smooth in wavelength (e.g., $\lambda^{-4}$-dependence for Rayleigh scattering) while molecular absorption often has structured spectra. In DOAS, the total spectral attenuation is therefore

split into a high-frequency part consisting of the (high-frequency components of) trace gas absorptions and a low-frequency part accounting for elastic scattering which is described by a low-order polynomial also compensating for intensity changes, e.g. caused by clouds. In addition, the effect of inelastic scattering, which is predominantly due to Rotational-Raman-Scattering known as the Ring effect (Shefov, 1959; Grainger and Ring, 1962) is accounted for by a pseudo cross section (e.g. Vountas

et al., 1998). Similarly, intensity offsets mostly resulting from stray light within the spectrometer are accounted for by pseudo cross sections $\sigma_{\mathrm{off}}$ (see Sect. 4.3 for more details). In total, the optical depth $\tau$ is approximated by

$$\tau = \ln\left(\frac{I_0}{I}\right) = \sum_i \sigma_i \cdot SC_i + \sigma_{\mathrm{Ring}} \cdot SC_{\mathrm{Ring}} + \sigma_{\mathrm{off}} \cdot SC_{\mathrm{off}} + \sum_p a_p \lambda^p + r \tag{1}$$

where the first sum is over all $i$ absorbers having cross section $\sigma_i$, the polynomial degree is $p$, and the residual term $r$ contains the remaining (uncompensated) optical depth. An important quantity

used within this study to identify and evaluate differences between DOAS retrieval codes is the root mean square of the fit residual $r$ (for simplicity denoted as RMS in the context of this study), which is a measure of the fit quality. In Eq. 1, known as the *DOAS equation*, all quantities with the exception of $a_p$ depend on wavelength $\lambda$. For tropospheric absorbers, the spectrum $I$ is normally taken at small elevation angles above the horizon where the tropospheric light path is large. The reference

spectrum $I_0$ is normally a zenith spectrum either at small solar zenith angle (SZA), which is in the following called a noon reference fit, or close in time to the measured spectrum $I$, in the following called a sequential reference fit.

The DOAS equation is usually an over-determined problem ($\tau$ is measured at more wavelengths $\lambda$ than unknowns exist in Eq. 1) and solved by means of a least-squares fit (see also Sect. 4.5), i.e.

minimizing the residual term. The resulting fit factors are the polynomial coefficients $a_p$ and the so-called slant columns $SC_i$, which are the quantities of interest. The slant column is the integrated concentration $\rho_i$ of absorber $i$ along the effective light path $s$ (for simplicity we use the SC also for the fit coefficients of the Ring and offset spectra):

$$SC_i = \int \rho_i ds \tag{2}$$

Note that this is a simplification as normally an ensemble of different light paths contribute to the

measurement. In extreme cases, the absorption optical depth can become a non-linear function of the trace gas concentration. However, this is usually only of importance for satellite limb measurements and is discussed in detail for example in (Pukite and Wagner, 2016). A comprehensive discussion of the DOAS technique can be found for example in (Hönninger et al., 2004; Platt and Stutz, 2008).

### 2.2 The MAD-CAT campaign

The Multi-Axis DOAS Comparison campaign for Aerosols and Trace gases (MAD-CAT) was carried out in Mainz, Germany, in summer 2013. During the intensive phase of the campaign (7 June to 6 July 2013), 11 groups deployed MAX-DOAS instruments on the roof of the MAX-Planck Institute for Chemistry at the Mainz University campus. Being located in the densely populated Rhine-Main region, the observations are dominated by anthropogenic pollution, predominantly by $NO_2$ (in the

visible). The main azimuthal viewing direction was 51° (from north), pointing roughly in the direction of the city of Frankfurt $\approx 30$ km away. Series of vertical scans comprising elevation angles of 0°, 1°, 2°, 3°, 4°, 5°, 6°, 8°, 10°, and 30° were performed. In each direction, single measurements (of varying





Table 1: Summary of fit settings used for the $NO_2$ intercomparison. These fit settings were agreed on during the MAD-CAT campaign and can be found also at `http://joseba.mpch-mainz.mpg.de/mad_analysis.htm`.

| Fit | Reference | Window | Cross sections | Intensity offset | Polynomial |
|-----|-----------|--------|----------------|------------------|------------|
| v1 | noon | 425-490 | 1,2,4,5,6,7 | Constant (0th order) | 5 (6 coefs) |
| v1a | sequential | 425-490 | 1,2,4,5,6,7 | Constant (0th order) | 5 (6 coefs) |
| v2 | noon | 411-445 | 1,3,4,5,6,7 | Constant (0th order) | 4 (5 coefs) |
| v2a | sequential | 411-445 | 1,3,4,5,6,7 | Constant (0th order) | 4 (5 coefs) |

Cross sections:
1   $NO_2$ at 298K (Vandaele et al., 1996), $I_0$-correction using 1E17 molec/cm$^2$
2   $NO_2$ at 220K orthogonalized to 298K within 425-490 nm
3   $NO_2$ at 220K orthogonalized to 298K within 411-445 nm
4   $O_3$ at 223K (Bogumil et al., 2003)
5   $O_4$ Hermans et al., unpublished, `http://spectrolab.aeronomie.be/o2.htm`
6   $H_2O$, HITEMP (Rothman et al., 2010)
7   Ring, NDSC2003 (Chance and Spurr, 1997)

exposure time depending on illumination) were integrated for 20 s (for the IUPB instrument, other instruments used different integration times). Between vertical scans, multiple zenith measurements were performed (see Fig. 1 for an example of resulting $NO_2$ slant columns). Detailed information about MAD-CAT can be also found at `http://joseba.mpch-mainz.mpg.de/mad_cat.htm` and the first campaign data focusing on range-resolved distributions of $NO_2$ measured at different wavelengths was published by Ortega et al. (2015). Furthermore, Lampel et al. (2015) demonstrated the presence of vibrational Raman scattering on $N_2$ molecules in spectra measured during MAD-CAT. In addition, publications based on MAD-CAT data are in preparation focusing on HCHO (Pinardi, 2017), HONO (Wang et al., 2016) and CHOCHO (Ortega et al., 2017).

### 2.3 The IUPB MAX-DOAS instrument

The IUPB MAX-DOAS instrument deployed in Mainz during the MAD-CAT campaign is a two-channel CCD-spectrometer system measuring in the UV and visible, respectively. Within this exercise, only data from the visible spectrometer are used as $NO_2$ is best retrieved in this spectral range. The spectrometer is an ANDOR Shamrock 303i covering a spectral range from 399-536 nm at a resolution of ≈0.7 nm. The spectrometer was actively temperature stabilized to 35°C. Spectra were recorded with a CCD of the ANDOR iDUS 420 type having 1024x255 pixels (26x26 μm each) leading to a spectral sampling of 7-8 pixels/nm.

Light was collected by a telescope unit mounted on a commercial ENEO VPT-501 pan-tilt head allowing pointing in any viewing direction. Photons entering the telescope through a fused silica window were focused by a lens on an optical fiber bundle. The instrument's field-of-view (FOV) was ≈ 1.2°. The Y-shaped optical fiber bundle (length 20 m) connecting the telescope to both spectrometers consists of 2x38 = 76 single fibres minimising polarization effects. A video camera inside the telescope housing takes snap shots for every recorded spectrum for scene documentation and a mercury-cadmium (HgCd) line lamp allows wavelength calibration measurements. Dark current and slit function measurements are performed every night. The same instrumental set up has been used in previous campaigns, e.g. CINDI and TransBrom (Roscoe et al., 2010; Peters et al., 2012).

### 2.4 Intercomparison exercise

Spectra recorded by the IUPB MAX-DOAS instrument on 18 June 2013 during MAD-CAT were distributed to partners. It is worth mentioning that this intercomparison exercise was not restricted to groups participating MAD-CAT, as common observations were provided. 18 June 2013 was selected as having good viewing and weather conditions. However, temperature stabilization of the IUPB





Table 2: List of participating institutes with abbreviations used within this study.

| Abbr. | Retrieval | Institute |
|---|---|---|
| IUPB | NLIN | Institute for Environmental Physics, University of Bremen, Germany |
| AUTH | QDOAS | Aristotle University of Thessaloniki, Greece |
| BIRA | QDOAS | Belgian Institute for Space Aeronomy, Brussels, Belgium |
| JAMSTEC | QDOAS | Japan Agency for Marine-Earth Science and Technology, Japan |
| Toronto | QDOAS | Department of Physics, University of Toronto, Ontario, Canada |
| IUPHD | DOASIS | Institute of Environmental Physics, University of Heidelberg, Germany |
| Boulder | QDOAS | University of Colorado, Boulder, USA |
| KNMI | KMDOAS | Royal Netherlands Meteorological Institute, De Bilt, The Netherlands |
| INTA | LANA | National Institute for Aerospace technology, Madrid, Spain |
| MPIC | MDOAS, WinDOAS | Max Planck Institute for Chemistry, Mainz, Germany |
| CSIC | QDOAS | Department of Atmospheric Chemistry and Climate, Institute of Physical Chemistry Rocasolano, CSIC, Madrid, Spain |
| NIWA | STRATO | National Institute of Water and Atmospheric Research, Lauder, New Zealand |
| IAP | RS.DOAS | A. M. Obukhov Institute of Atmospheric Physics, Russian Academy of Sciences, Moscow, Russia |
| BSU | WinDOAS | Belarusian State University, Minsk, Belarus |
| USTC | QDOAS | University of Science and Technology, Hefei, China |
| UNAM | QDOAS | National Autonomous University of Mexico, Mexico |
| NUST | QDOAS | Institute of environmental sciences and engineering (IESE), National University of Sciences and Technology, Islamabad, Pakistan |

spectrometer was problematic due to unusually hot weather which due to a lack of air conditioning on that day led to overheating of the instrument. As a result, spectral stability was not as good as usual, providing the opportunity to investigate how different retrieval codes deal with potential spectral shifts during the day. The provided data comprised observations at several elevation angles in the main azimuthal viewing direction of 51° (see Fig. 1 for an example). In addition, common trace gas cross sections as well as DOAS fit settings were distributed as summarized in Tab.1, which are the
common fit settings as agreed on during the MAD-CAT campaign. Note again, the objective of the present study is not finding optimal $NO_2$ fit settings but identifying disagreements between retrieval codes. All groups then performed DOAS fits using these harmonized settings on IUPB spectra using their own retrieval software. The influence of different retrieval codes was then analysed, focusing on the impact on $NO_2$ columns. In the following, a brief description of retrieval codes used by the
different groups is given. Participating groups/institutes and corresponding abbreviations used within this study are summarized in Tab.2.

### 2.4.1 IUPB

The IUPB retrieval code NLIN (Richter, 1997) is an in-house software originally developed for the analysis of ground-based measurements but over time extended and also used for airborne and satellite
data sets. The DOAS matrix is inverted using a singular value decomposition (SVD, see Sect. 4.5) which is comprised in an iterative nonlinear fit for calibration of the wavelength axis using a Marquardt-Levenberg fit (see Sect. 4.6). Comparisons with other retrieval codes have repeatedly demonstrated excellent agreement.

### 2.4.2 QDOAS users

The QDOAS software (Dankaert et al., 2013) is the multi-platform (Windows, Unix/Linux and Mac) successor of the WinDOAS software (Fayt and Van Roozendael, 2001) developed since the 1990s at





BIRA for the analysis of DOAS applications. QDOAS uses a coupled linear/non-linear least squares (NLLS) algorithm (Marquardt-Levenberg fitting and SVD decomposition) to solve the DOAS equation and includes a wavelength calibration module where measured intensities are fitted to a high resolution reference solar spectrum. During this operation, the slit function can be characterized in addition to the wavelength registration of the measured spectra.

QDOAS is under constant evolution to match the needs of developing ground-based, aircraft and satellite applications. It is widely used within the DOAS community (more than 130 institutes in 42 countries) and has been applied in different versions for the present exercise: AUTH (version 2.109.3), JAMSTEC (version 2.109), University of Toronto (version 2.109), CSIC (QDOAS windows 2.110.1 beta 20151123), BIRA (version 2.110.1), USTC (version 2.109.4), UNAM (version 2.109.3), NUST (version 2.111), CU Boulder (version 2.110). These various versions do not differ in the core-implementation of the DOAS analysis, but can feature slightly different capabilities in terms of data handling or specific retrieval functionalities (see `http://uv-vis.aeronomie.be/software/QDOAS/LastChanges.php`).

### 2.4.3 INTA

The INTA retrieval software is named LANA (actual version number: v7.0.0). LANA has been developed at INTA and used since 1994. It is a two-step iterative algorithm. In the first step, cross section and spectra are positioned and in the second step the linear equations system is solved using a Gauss-Jordan procedure. INTA DOAS instruments using LANA software have participated for example at the CINDI campaign in 2009 (Roscoe et al., 2010).

### 2.4.4 MPIC

MPIC uses different retrieval codes and procedures, which are included in this intercomparison. In MPIC_WD, the WinDOAS software has been used. This analysis was performed for noon reference fits only and the reference has been selected manually. Another analysis referred to as MPIC_MD has been performed using the retrieval code MDOAS written in Matlab (but calibration and convolution have been performed in WinDOAS). For sequential reference fits, two versions MPIC_MDa and MPIC_MDb exist, which used a different treatment of the Ring spectrum.

### 2.4.5 KNMI

The KNMI retrieval code KMDOAS was developed in 2013-2014 at KNMI (by A. Piters) and verified using QDOAS. It is written in Python using standard modules (matplotlib, numpy, scipy, pandas).

### 2.4.6 IUPHD

IUPHD is using the DOASIS software in its version 3.2.4595.39926 (Kraus, 2006).

### 2.4.7 NIWA

The *NIWA-Strato* package of programs roots in the first DOAS days (1980s). It was originally used for processing zenith DOAS spectra, but extended as needed to handle MAX-DOAS measurements. The fitting code uses least squares or optional SVD fitting inside iterations applying shift, stretch and (optional) offset, for minimum residual. An internal equidistant wavelength grid is applied using the average interpixel spacing.

### 2.4.8 IAP

The IAP group uses an own-developed software RS.DOAS for both, wavelength calibration procedure and slant column retrieval. The IAP algorithm was described in (Ivanov et al., 2012) and the more recent version in (Borovski et al., 2014) (in application to formaldehyde retrieval). Most recent developments of the IAP algorithms were described in (Postylyakov et al., 2014) and (Postylyakov and Borovski, 2016). For convolution of cross-sections the QDOAS software (offline version 2.0 at 5 March 2012) is used.





In the IAP algorithm the wavelength calibration of the reference spectrum (WCRS) procedure is iterative. The WCRS algorithm uses several subwindows, and for each of them a non-linear shift and stretch fit including all trace gases (resulting from the linear DOAS fit) is applied. From a set of shifts obtained for the subwindows a 2nd-order polynomial approximation of the shift dependency on pixel number is constructed. The WCRS is corrected using values of this polynomial. The corrected calibration is used as input for the next iteration. This process is repeated until corrections of wavelength calibration become less than 0.001 nm.

The IAP algorithm of the spectra analysis for obtaining SC is similar to WCRS one, but an intensity offset is added to the non-linear terms of the DOAS retrieval procedure and one window is used. Each analyzed spectrum is interpolated to the calibrated grid of the reference spectrum using the 3rd order Lagrange polynomial (Postylyakov and Borovski, 2016). The subroutine of LU decomposition is used for matrix inversions in the algorithm (ALGLIB software, Sergey Bochkanov, `www.alglib.net`).

### 2.4.9 BSU

BSU has used WinDOAS (v.2.1, 2001) for this intercomparison exercise.

### 2.4.10 Boulder

CU-Boulder used QDOAS v2.110 for wavelength calibration, convolution of trace gas cross sections, and slant column retrieval. The primary zenith DOAS reference was calibrated iteratively against a Kurucz spectrum fitting Ring, $O_3$, and $NO_2$ in several subwindows with the fitted shift of the sub-windows used to generate a 3rd order polynomial of shift across the detector. For changing reference analyses the secondary zenith references were calibrated (using the primary reference calibration a priori), but the procedure was run only once. The reference used for changing reference analyses was the zenith immediately following the elevation angle scan analyzed. High resolution cross-sections were convolved online following each calibration, and all cross-sections and the reference were allowed to shift and stretch (1st order) relative to the spectrum.

## 3 Intercomparison results

### 3.1 Noon reference, 425-490 nm fit window (v1 fit parameters)

Differences between groups for the 425-490 nm fit using a noon reference (v1 fit settings, see Tab. 1) are shown in Fig. 3 for individual measurements at an elevation angle of 2°. Small elevation angles above the horizon are associated with long tropospheric light paths and therefore important for the detection of tropospheric absorbers. Differences shown in Fig. 3 are relative to IUPB results. For the objective of identifying retrieval-code specific effects, the use of a single retrieval code as a reference seems advantageous in comparison to using the mean of all retrieval codes which would average over all retrieval-specific features. Note that this does not exclude IUPB from the intercomparison as problems of the IUPB retrieval would be easily detected as leading to the same systematic patterns in all lines shown in Fig.3.

Absolute differences (institute-IUPB) and relative differences (absolute difference/IUPB) of $NO_2$ slant columns and fit RMS are shown in Fig. 3a-d. In general, $NO_2$ slant column differences are in the range of $\pm 2 - 3 \cdot 10^{15}$ molec/cm$^2$ or $< 2\%$. This is about a factor of 2-3 larger than $NO_2$ slant column errors, which are typically $< 1 \cdot 10^{15}$ molec/cm$^2$, resp. $< 0.6\%$ for 2° elevation. A clearly enhanced disagreement is observed for the first data point in the INTA time-series as well as for MPIC_WD. The latter could be linked to the reference spectrum, which is in this case not the one having smallest sun zenith angle (SZA) while the outlier in the INTA timeseries was identified to arise from different implementations of the intensity offset correction (see Sect. 4.3). Note that these $NO_2$ differences for individual measurements are much smaller than the variability (diurnal cycle) of $NO_2$ and thus almost invisible in Fig. 3e where no differences but absolute $NO_2$ slant columns from each groups (including IUPB in black) are plotted.





Figure 3: Results from v1 fit settings in 2° elevation angle as a function of time. (a, b) Absolute NO$_2$ slant column and RMS differences, (c, d) relative NO$_2$ slant column and RMS differences, (e) NO$_2$ slant columns, (f) RMS, (g, h) fitted spectral shift (h is a zoom-in of g without INTA and KNMI lines). (a-d) are differences w.r.t. IUPB, (e-h) are absolute results and IUPB is explicitly shown in black.




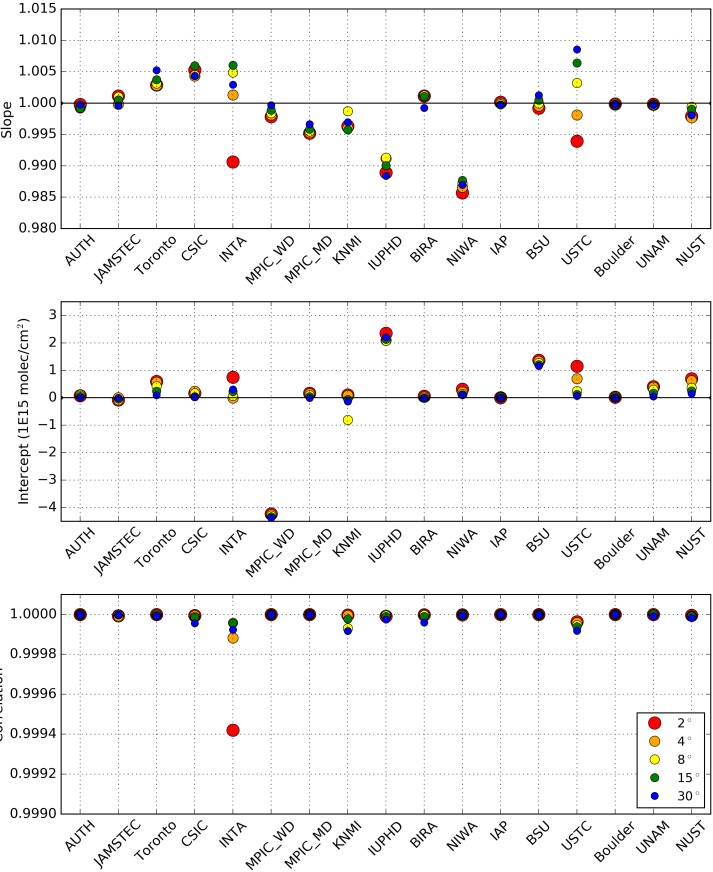

Figure 4: Linear regression results (slope, intercept, correlation) for different elevation angles (w.r.t.
IUPB) for fit settings v1 (noon reference).

Interestingly, most groups show a smooth behavior (constant offset) in absolute $NO_2$ differences
(Fig. 3a), which is mostly an effect of the choice of the reference (see Sect. 4.1) while relative differences
(Fig. 3c) reflect the shape of $NO_2$ slant columns in Fig. 3e (smaller slant columns lead to larger relative
differences and vice versa). However, some groups show a smooth line not for absolute, but for relative
differences, e.g. NIWA. Thus, two types of disagreements are observed, 1) constant in absolute, and
2) constant in relative differences. These two types are linked to differences in retrieval codes, which
is investigated in detail in Sect. 4.

Absolute RMS differences (w.r.t. IUPB) in Fig. 3b show the same shape as $NO_2$ slant columns.
This is because at small elevations the RMS itself reflects the shape of the $NO_2$, which was already
demonstrated and discussed in Fig. 1. A better measure for the identification of differences between
retrieval codes is thus the relative RMS disagreement shown in Fig. 3d. Interestingly, the first data
point for INTA showing a large disagreement with IUPB and other groups in $NO_2$ slant columns is
prominent in absolute RMS differences as well, but not in relative RMS differences. The reason is that
the RMS in the morning is very large (Fig. 3f, compare also to Fig. 1) and thus decreases the relative
difference. Remarkably, relative RMS differences are found up to 80-100%, which is substantially more
than $NO_2$ slant column differences (only a few percent). In addition, some clusters can be seen in
Fig. 3d: A group of smallest RMS comprising e.g. IUPB, BIRA, CU Boulder, AUTH, IAP, and a
group of slightly enhanced RMS ($\approx 20\%$) comprising e.g., MPIC_WD, NIWA, BSU. In the group of



Table 3: Ranges of correlations, slopes and offsets from linear regressions on $NO_2$ slant column correlation plots between each group and IUPB (see also Figs. 4 and 6).

| Fit | Correlation (%) | Slope | Offset ($1 \cdot 10^{15}$molec/cm$^2$) |
|-----|-----------------|-------|----------------------------------------|
| v1 | >99.98 (without outlier) | 0.985 to 1.01 | -4 to 3 |
| v1a | >99.2 | 0.96 to 1.01 | -1.5 to 1 |
| v2 | >99.94 (without outlier) | 0.985 to 1.005 | -4 to 3 |
| v2a | >99.2 | 0.96 to 1.01 | -2 to 1 |

largest RMS (up to 80-100%), only UNAM and NUST show similar features, while the timeseries of INTA and USTC is different.

The IUPB spectra provided were wavelength pre-calibrated using nightly HgCd line lamp measurements as explained in Sect. 2.3. However, the DOAS fit quality can be improved (RMS reduced) by applying a post-calibration. In addition, the nighttime calibration can change during the day as a
result of temperature drifts. This is accounted for by DOAS retrieval codes in terms of a nonlinear shift fit (for a more detailed discussion see Sect. 4.6). Figs. 3g and h (being a zoom-in of g) show the reported shift resulting from the wavelength calibration in participating retrieval codes. Note, absolute values for the shift are shown here and the IUPB result is explicitly included. For an ideal spectrometer without drifts, only a very small shift would be expected caused by a non-commutivity
of convolution and DOAS polynomial, which is known as the *tilt effect* and typically in the order of less than 1-2pm, depending on the instrument resolution (Sioris et al., 2003; Lampel et al., 2016). The shift retrieved here is larger than that and driven by overheating of the system on this day. Timeseries of shifts shown in Fig. 3h agree well in shape. A small offset is observed for NIWA, which could be caused by NIWA not centering the slit function accurately in the making of their cross sections, but
this does not affect their results. Also, the intensity offset calculation was previously implemented in a different way in the NIWA retrieval and changed for this exercise to meet the common fit settings, but this new code had a fault which was discovered after data submission and successfully corrected (the corrected results are not shown here). In contrast, as shown in Fig. 3g, KNMI is fitting a rather different shift (and provides the shift only in 0.01 nm resolution which leads to the displayed discrete
steps). This is caused by different definitions of the shift (see Sect. 4.6): KNMI is fitting the shift of the optical depth relative to the cross sections while the other groups are fitting the shift of $I$ relative to the reference spectrum $I_0$. As a result, the shift of KNMI and other groups cannot be expected to match. Apart from KNMI, only INTA is retrieving clearly different shifts. The effect of the wavelength calibration is investigated in more detail in Sect. 4.6.
To further quantify the agreement between groups and include also other elevation angles, correlation plots were calculated for each group w.r.t. IUPB $NO_2$ slant columns (not shown) and a linear regression was performed for each plot providing a slope, offset, and the correlation coefficient. The results are summarized in Fig. 4, color-coded for different elevation angles. As expected, the correlation coefficient is $> 99.98\%$ with the only exception of the INTA 2° elevation which is predominantly caused by
the outlier already seen in Fig. 3. The slope ranges between 0.995 and 1.01, the offset between -4 to $2.5\cdot10^{15}$ molec/cm$^2$. Apart from USTC and INTA, no large separation of slope and offset with elevation angle is observed. An important observation is that groups using the same retrieval code (QDOAS) do not necessarily show the same systematic behavior in Fig. 4, implying that the influence of remaining fit parameters different from the harmonized general settings in Tab. 1 is still larger than
the effect of the specific retrieval software used. In general, the best agreement in terms of regression line slope, offset and correlation coefficient is found between IUPB, AUTH, IAP, and CU Boulder, which are all using different retrieval codes.

## 3.2 Other fit parameters

The agreement between groups was quantified in the same way using a sequential reference (see
Sect. 2.1) instead of a noon reference spectrum, which is often preferred if tropospheric absorbers are of interest as stratospheric effects are removed to a large extent.





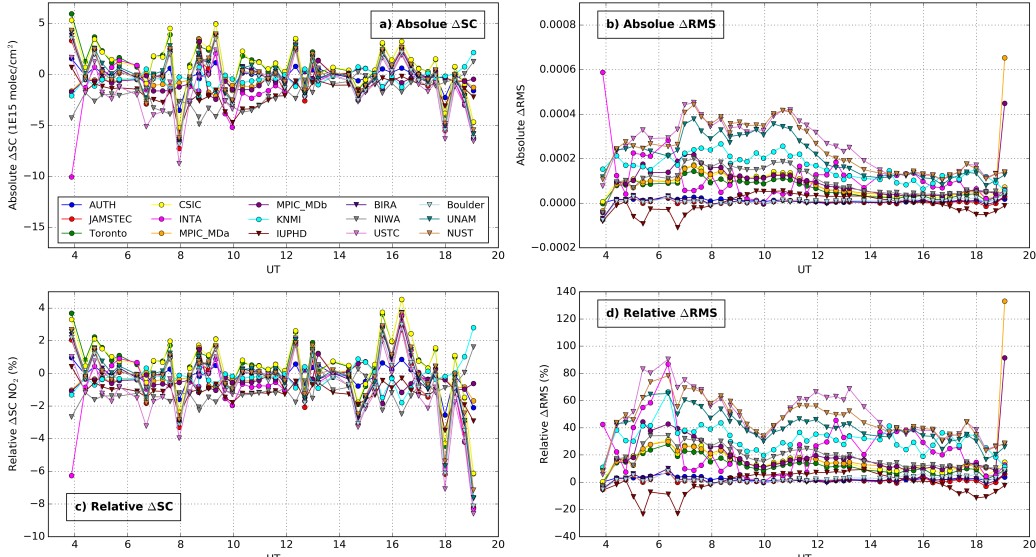

Figure 5: Results from v1a fit settings (sequential reference spectrum) on 2° elevation angle data as a function of time. (a and b) Absolute $NO_2$ slant column and RMS differences, (c and d) relative $NO_2$ slant column and RMS differences. All differences are relative to IUPB.

Figs. 5 and 6 are similar to Figs. 3 and 4 but using a sequential reference, denoted as v1a fit settings instead of v1 (see Tab. 1). Note that some groups are missing here (e.g. IAP) as they only provided noon reference fits. The range of disagreements for individual $NO_2$ slant columns is up to 8% and

therefore larger than $NO_2$ disagreements using a noon reference (v1 fit). In addition, neither absolute nor relative differences between groups are smooth (Fig. 5 a and c). This is because in contrast to the noon reference, a different (zenith) $I_0$ spectrum is used for every scan and thus the impact of details of the implementation of the reference changes from scan to scan. Different implementations of the reference spectrum are further investigated in Sect. 4.1 and were found to be the major reason for

$NO_2$ disagreements between groups. The outlier (first data point) seen for v1 (noon) fit settings is present for v1a as well but it is not prominent here as fluctuations for the above mentioned reason are of same magnitude.

In terms of RMS, differences between groups are comparable to v1 results and as large as 80%.

Compared to the noon reference fit (Fig. 4), correlations from linear regressions in Fig. 6 are smaller,

especially for the 30° elevation. This results from the larger disagreements as explained above, but is also partly expected as slant columns are smaller using a sequential reference. Correlations are still > 99.2% for 30° elevation and even > 99.8% for smaller elevations. The intercept is mostly below ±1E15 molec/cm² and therefore smaller than in the v1 fit. In contrast, the slope of regression lines ranges between 0.96 and 1.01 and is therefore more variable than in v1. Furthermore, in contrast to

v1 where no systematic pattern was observed, all groups using QDOAS (with the exception of USTC) show the same pattern in terms of slope (largest for 2°, smallest for 30°) and even a comparable range, while groups using independent software (INTA, MPIC_MD, KNMI, IUPBHD, NIWA) do not show this systematic pattern. The reason is that until version v.2.111 only one possibility for the sequential reference was implemented in QDOAS (the closest spectrum after the scan, with v.2.111

other options have been included). Consequently, the sequential reference selection is applied in the same way in all QDOAS data sets shown here. This is consistent with findings of Fig. 5 where the exact implementation of the sequential reference was already found to dominate $NO_2$ differences between groups.

In addition to the 425-490 nm fit (v1 and v1a), a smaller fit window used within the MAD-CAT





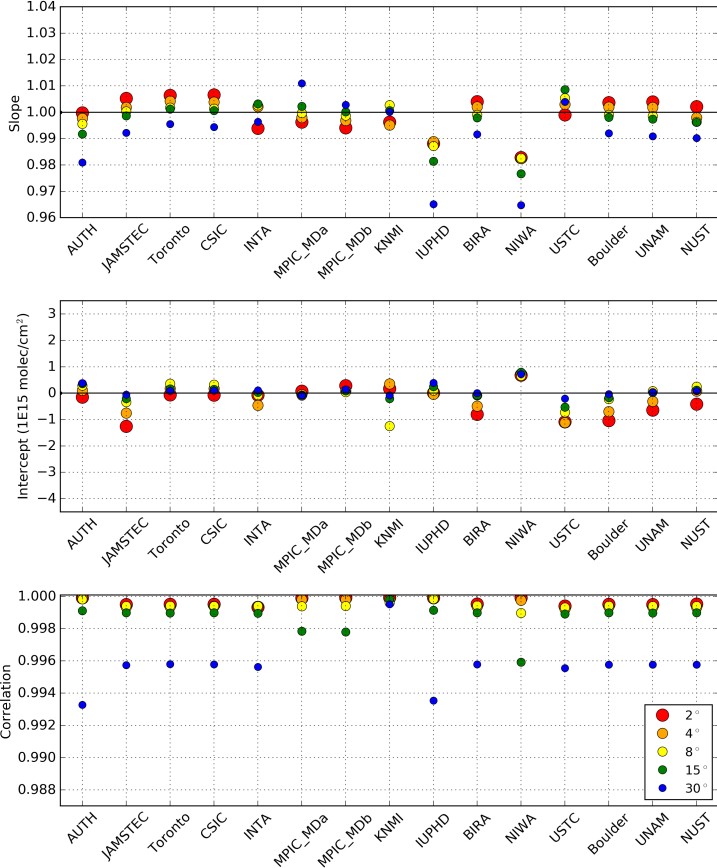

Figure 6: Linear regression results (slope, intercept, correlation) for different elevation angles (w.r.t. IUPB) for fit settings v1a (sequential reference).

campaign was intercompared (v2 and v2a using a spectral range of 411-445 nm, see Tab. 1). However, results of fit settings v2 and v2a are not explicitly shown as providing no new insights but confirming observations and findings above. Typical values of statistics (from linear regressions) are summarized in Tab. 3 for all performed fit settings. A very small tendency of better agreement between groups if using the larger fit window is seen. Although this should not be over-interpreted, it could be caused by more information being present in the large fit window and therefore more accurate results (less statistical fluctuations).

## 4 Understanding differences between retrieval codes

Questionnaires sent to every group suggested five potential sources of differences in the DOAS fit results: 1) the selection/calculation of the reference spectrum, predominantly for sequential reference fits, 2) treatment of the slit function, 3) the intensity offset correction, 4) differences in the numerical calculation of the DOAS fit (linear fit), 5) differences in the wavelength calibration (non-linear fit). In the following, tests using the same retrieval code (IUPB) have been performed to characterize the impact of each of the above mentioned systematic differences.





Table 4: Tests performed to study the influence of different reference spectra.

| Test | Reference (fit setting) | Remarks |
|------|------------------------|---------|
| TR0 | noon (v1) | first spectrum of smallest SZA |
| TR1 | noon (v1) | second spectrum of smallest SZA |
| TR2 | noon (v1) | first for a.m., second for p.m. |
| TR3 | noon (v1) | average of both |
| TR4 | sequential (v1a) | closest zenith in time |
| TR5 | sequential (v1a) | closest before the scan |
| TR6 | sequential (v1a) | closest after the scan |
| TR7 | sequential (v1a) | average of before and after |
| TR8 | sequential (v1a) | interpolation of before and after to time of measurement |

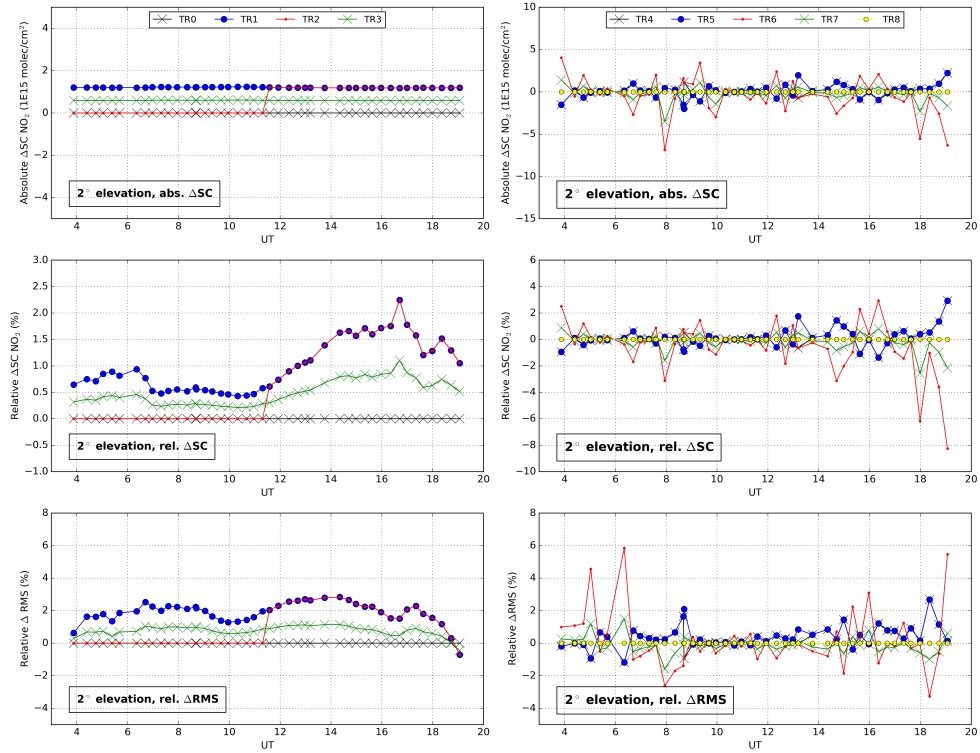

Figure 7: Different test results (see Tab. 4) in 2° elevation angle as a function of time. Top: Absolute $NO_2$ SC differences, Middle: Relative SC differences, Bottom: Relative RMS differences. Left is for noon reference (differences are w.r.t. IUPB v1 fit results), right is for sequential references (differences are w.r.t. IUPB v1a fit results).

## 4.1 Effect of the reference spectrum

In the IUPB spectra provided to intercomparison partners, two different zenith spectra at the same smallest SZA were reported, which is of course non-physical. Actually, the second zenith spectrum is the one having the smallest SZA, but for rounding reasons (the SZA was a 4-digits number in the spectra file provided), both spectra had the same SZA. Consequently, for the noon reference fits v1 and v2 within this intercomparison exercise, four options exist: 1) taking the first zenith spectrum





of smallest SZA, 2) taking the second one, 3) taking the first for a.m. and the second for p.m. (i.e. always taking the closest in time), and 4) taking the average of both spectra. Similarly, different options exist for calculating the sequential references for fits v1a and v2a: 1) always taking the zenith spectrum closest in time, 2) always taking the last zenith spectrum before the actual measurement, 3) always taking the next zenith spectrum after the measurement, 4) taking the average of the two

(before and after), and 5) interpolating the two zenith spectra to the time of the actual measurement.

All different options for noon and sequential reference fits were evaluated using the IUPB retrieval code NLIN (Tab. 4). Fig. 7 shows the resulting absolute and relative slant column differences (top and middle) as well as relative RMS differences (bottom) for noon reference (left) and sequential reference (right) w.r.t. v1, resp. v1a fit results.

Taking another noon reference spectrum results in a constant offset in absolute $NO_2$ differences (Fig. 7, top left). Test TR0 (using the first spectrum) yields the same results as the IUPB v1 fit from Sect. 3, because there the first zenith spectrum was used as a reference as well. In contrast, using the second zenith spectrum as a reference (TR1) results in a constant offset of 1.5E15 molec/cm$^2$ (0.5-2.5% in relative differences, depending on the actual $NO_2$ slant column), because the second zenith

spectrum had apparently a smaller $NO_2$ content. The change of the $NO_2$ content could be related to changes of the atmospheric $NO_2$ amount or the atmospheric light path. In terms of RMS, TR1 is up to 3% larger (Fig. 7, bottom left). The main reason for this is probably the larger $NO_2$ slant columns as associated effects like the wavelength-dependence of the $NO_2$ slant column (Pukite et al., 2010) were not compensated in this intercomparison exercise as discussed in Sect. 1.

Test TR2 yields results which are identical to TR0 am and TR1 pm values. This is not seen for any groups in Fig. 3 above, i.e. this option is apparently not present in any retrieval code. Not surprisingly, TR3 (averaging both zenith spectra) yields results which are between TR0 and TR2.

In contrast to noon reference tests, sequential references show no smooth behavior, neither for absolute, nor for relative differences (Fig. 7, right). The reason is that for each vertical scanning

sequence (from which only the 2° elevation is shown here and in Fig. 3) another reference spectrum is used and consequently reference-related differences between groups are also changing from scan to scan. Note that almost no difference can be seen in Fig. 7 between TR4 and TR5 as the 2° elevation angle is shown here and consequently the closest zenith measurement in time is normally the one before the scan as IUPB measurements proceed from low to large elevations. TR8 (interpolation to

the measurement time) resembles the sequential reference treatment normally implemented in the IUPB code, and thus the TR8 line is zero. For TR6 and TR7, absolute and relative differences (up to 8%) are remarkably similar to observed differences between groups using v1a fit settings, both in shape and in absolute values (compare to Fig. 5).

To conclude, the exact treatment of reference spectra is the major reason for observed $NO_2$ dis-

crepancies between groups in Sect. 3, causing differences of up to 8%. Unfortunately, no clear recommendation can be derived from relative RMS differences in Fig. 7 (bottom right) as all lines scatter around zero. However, the relative difference in RMS can be as large as 6% and in general, using a single zenith reference spectrum before or after the scan tends to produce larger RMS. However, while the reference treatment explains the majority of $NO_2$ disagreements, it cannot explain the large RMS

differences (up to 100%) between groups.

## 4.2 Slit function treatment

The slit function distributed to intercomparison participants originated from HgCd line lamp measurements made in the night before 18 June 2013. It was pre-processed in terms of subtraction of the (also measured) dark signal, centered, and provided on an equidistant 0.1 nm grid.

While some groups/retrieval codes used this slit function as is, other retrieval codes include a further *online* processing of the slit function, e.g. by fitting line parameters. In order to quantify the effects of this processing on the resulting slant columns, cross sections have been convolved off-line (before the fit) using different treatments of the slit function as summarized in Tab. 5. Then, again, the IUPB retrieval code NLIN has been used to calculate $NO_2$ slant columns.

Examples of different treatments of the slit function are shown in Fig. 8. The original slit function is displayed in blue. If fitting a Gaussian shape to it, the maximum is not exactly centered around zero.





Table 5: Different treatments of the slit function.

| Test | sub. offset | geom. centering | cutoff value | fit line parameters |
|------|-------------|-----------------|--------------|---------------------|
| TS0  | no  | yes | 0.0   | no |
| TS1  | no  | no  | 0.001 | no |
| TS2  | yes | no  | 0.001 | no |
| TS3  | no  | yes | 0.001 | no |
| TS4  | yes | yes | 0.001 | no |
| TS5  | no  | no  | 0.0   | no |
| TS6  | no  | no  | 0.001 | Gaussian |
| TS7  | no  | no  | 0.001 | Gaussian ($\mu = 0$) |
| TS8  | no  | no  | 0.001 | Gaussian ($\mu = 0$) fitted on 0.01 nm grid |
| TS9  | no  | no  | 0.0   | Gaussian |
| TS10 | no  | no  | 0.0   | Gaussian ($\mu = 0$) |
| TS11 | no  | no  | 0.0   | Gaussian ($\mu = 0$) fitted on 0.01 nm grid |

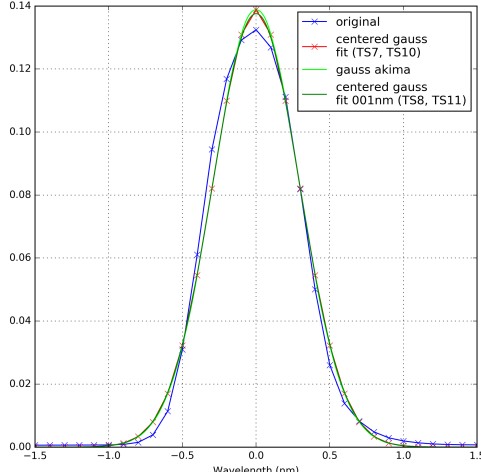

Figure 8: Examples of different slit function treatments. Blue: Measured slit function (from HgCd
line at $\approx 480$ nm). Red: Fitting a Gaussian shape to slit function with $\mu = 0$. Light green:
Akima interpolation of the red line to a finer 0.01 nm grid. Dark green: Interpolating first
(linearly) to 0.01 nm and fitting a Gaussian shape afterwards.

The slit function after fitting a Gaussian shape and forcing centering (i.e. $\mu = 0$) is shown in red (as
used in TS7 and TS10). Furthermore, performing a discrete convolution of cross sections requires the
same wavelength sampling, i.e. the slit function has to be interpolated to the cross section grid, which
was 0.01 nm. Here, an Akima interpolation has been applied (green line). However, if the original slit
function is first linearly interpolated to the required 0.01 nm grid and then a Gaussian shape is fitted,
a slightly different result is obtained, shown in dark green (as used in TS8 and TS11). Note that often
not a discrete convolution but a (faster) convolution using Fourier transformation is implemented in
DOAS retrieval codes (see below).

Fig. 9 shows the resulting absolute and relative differences in NO$_2$ slant columns (top and middle)
as well as relative differences of the fit RMS (bottom) w.r.t. the IUPB v1 fit (without I0-correction as
this was not applied to the slit function test fits). The reference IUPB v1 fit uses an online convolution
of cross sections using FFT and a further geometrical centering of the slit function is applied.

No difference between the reference fit and TS0 and TS5 results (black and magenta lines) is



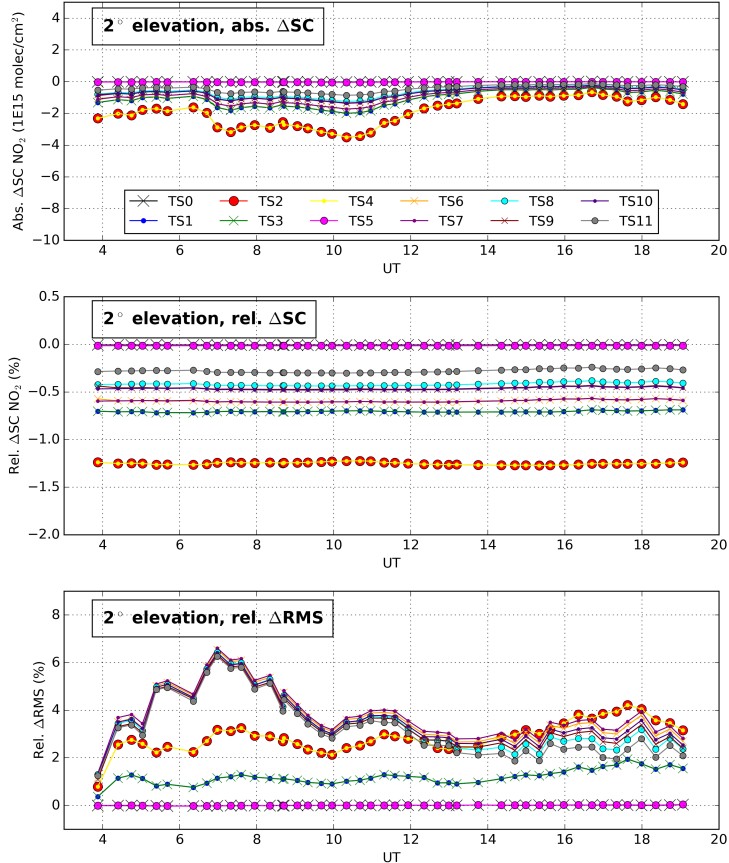

Figure 9: NO$_2$ slant columns and RMS differences w.r.t IUPB v1 fit results in 2° elevation angle for different treatments of the slit function.

observed, neither for NO$_2$ nor for the RMS. In TS0, the same slit function treatment is applied as in the reference fit, but using discrete convolution instead of FFT. Consequently, no difference arises due to the method of convolution. In addition, TS5 and TS0 are identical tests except for TS0 applying an explicit geometrical centering (i.e. centering the area) of the slit function (overcoming potential small deficits in centering during the pre-processing). As this has no visible effect on NO$_2$ slant columns

and RMS, the centering during pre-processing was already sufficient and the fact that the maximum is not exactly located at zero when fitting a Gaussian shape must be caused by the shape of the original line.

The largest impact on NO$_2$ in Fig. 9 (up to 1.3% smaller NO$_2$ slant columns) is seen for TS2 and TS4 which both subtract the smallest value from the slit function (i.e. forcing the slit function to be

zero for the smallest value measured). This is usually not performed in DOAS fits and only advisable if instrumental stray light is a large problem or if the dark signal drifts which is normally not the case in state-of-the-art CCD detectors that are cooled and temperature-stabilised using Peltier elements.

In contrast, the largest effect in terms of RMS (up to 6.5% larger RMS, Fig. 9 bottom) occurs not for TS2 and TS4, but for all tests using a fitted Gaussian instead of the original slit function.

Interestingly, the Gaussian tests (TS6-TS11) show almost no difference among them in terms of RMS in the morning, but split up towards the evening. This might have to do with decreasing NO$_2$ slant columns in the afternoon or changes of the slit function during the day. In terms of NO$_2$, all tests



Table 6: Tests performed for different implementations of the intensity offset correction (and $I_0$ correction).

| Test | Offset order | Offset approach | $I_0$-correction |
|------|--------------|-----------------|------------------|
| TI0 | 0th (constant) | 1/I (simple approach) | yes |
| TI1 | 0th (constant) | $1/I_0$ (simple approach) | yes |
| TI2 | 0th (constant) | Eq. 4 using $I$ (more sophisticated) | yes |
| TI3 | 0th (constant) | Eq. 4 using $I_0$ (more sophisticated) | yes |
| TI4 | None | No offset correction | yes |
| TI5 | 1st (constant + slope) | $1/I$, $\lambda/I$ (simple approach) | yes |
| TI6 | 0th (constant) | $1/I$, additionally Ring·$\lambda$ | yes |
| TI7 | 0th (constant) | 1/I (simple approach) | no |

using Gaussian slit functions yield smaller $NO_2$ slant columns of 0.2-0.7%.

An important finding is that all tests of different slit function treatments lead to constant offsets (smooth lines) in relative differences (and not in absolute differences). In addition, all tests yield larger RMS than the reference fit, i.e. using the measured slit function yields the lowest residuals.

### 4.3 Intensity offset correction

Photons may hit the CCD detector at locations not corresponding to their wavelength (e.g. through scattering on mirrors, surfaces etc. inside the spectrometer) which produces an intensity offset, also called stray light. In addition, other effects such as changes in the dark current can lead to intensity offsets and the vibrational Raman scattering (VRS) is known to produce spectral effects that are very similar to intensity offsets (Peters et al., 2014; Lampel et al., 2015). In the DOAS fit, usually pseudo cross sections are included in order to compensate for intensity offsets. If the measured spectrum $I$ is superimposed by a constant intensity $C$ (which is the most simple assumption), the optical depth reads

$$-\tau = \ln\left(\frac{I+C}{I_0}\right) = \ln\left(\frac{I}{I_0}\right) + \ln\left(1 + \frac{C}{I}\right) \approx \ln\left(\frac{I}{I_0}\right) + \frac{C}{I} \qquad (3)$$

with the Taylor expansion $\ln(1+x) \approx x$. Thus, in first approximation the intensity offset causes an additive term of optical depth that is proportional to $1/I$, which is often used as a pseudo absorber (and showing large similarities to the Ring cross section as this compensates a filling-in of Fraunhofer lines).

However, often more sophisticated approaches are used. For example, the IUPB retrieval code NLIN allows either the simple implementation of $\sigma_{\text{off}} = 1/I$ or

$$\sigma_{\text{off}} = \ln\left(\frac{I + C \cdot I_{\text{max}}}{I}\right) \qquad (4)$$

omitting the Taylor expansion in Eq. 3 and superimposing $I$ by a certain constant $C$ of the maximum intensity within the fit interval. In addition, sometimes also higher correction terms are used assuming that not only a constant superimposes the spectrum, but also a contribution changing with wavelength (in which case often $\lambda/I$ is included in addition to the simple approach $1/I$ in the DOAS fit). Furthermore, sometimes the offset is not included in the linear DOAS fit, but fitted nonlinearly (this is not included in tests performed here).

Different implementations summarized in Tab. 6 were tested in order to evaluate the influence of the intensity offset correction. Fig. 10 shows resulting absolute and relative differences of $NO_2$ slant columns and RMS values. Again, the reference for these differences is the IUPB v1 (noon) fit.

No difference in $NO_2$ or RMS is seen between TI0 and the v1 fit as both fits use the same simple approach of $\sigma_{\text{off}} = 1/I$. In contrast, TI1 uses $1/I_0$. In terms of $NO_2$ differences, the TI1 line follows slightly the shape of total fit RMS and $NO_2$ slant columns (compare to Fig. 3). Interestingly, while the disagreement of TI1 w.r.t the reference fit is on average $\approx 2\%$ for $NO_2$, the first data point is





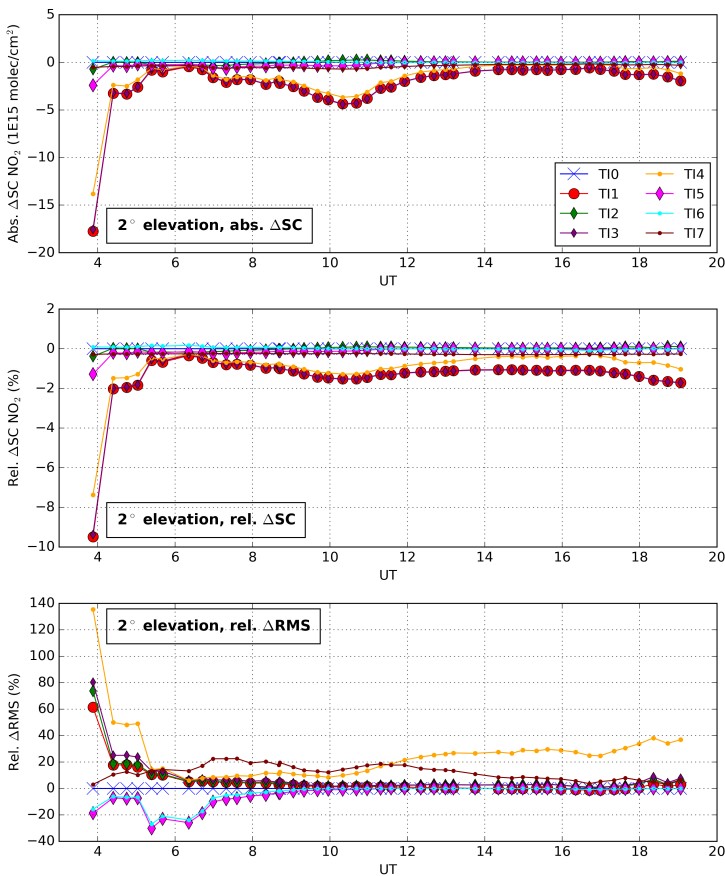

Figure 10: Absolute (top) and relative (middle) NO$_2$ slant column differences, and relative RMS differences (bottom) w.r.t. IUPB v1 fit results in 2° elevation angle resulting from different implementations of the intensity offset correction (and I$_0$ correction).

clearly off by almost 10% for NO$_2$ and 60% for RMS. This agrees perfectly with the observed outlier in the INTA analysis (compare to Fig. 3), i.e. the reason for this disagreement could be identified as a different offset implementation (which has been verified by INTA). The SZA and the sun azimuth angle (SAA) of this measurement are ≈90° and 58° (from north), respectively, while the instrument's elevation angle is 2° and the azimuthal viewing direction 51°, i.e. the instrument was pointing close to the rising sun. Enhanced stray light in the spectrometer (caused by the large contribution of photons at longer wavelengths while observing the red sky during sunrise) seems plausible and using 1/I is a better choice for compensation. It was verified (not shown) that the fit coefficient of the offset (also called the *offset slant column*) is particularly large not only in this but also in adjacent measurements and a color index indicated that these spectra are indeed more reddish. However, these measurements could also be affected by direct sunlight in the instrument, which is known to increase RMS (e.g. due to polarization issues). Fig. 3f demonstrates that the respective measurement is affected by a very large RMS (potentially caused by a combination of direct light and stray light).

TI2 and TI3 are more sophisticated approaches (Eq. 4) based on either I or I$_0$. However, the resulting lines follow largely TI0 and TI1 (simple approaches). Unexpectedly, both TI2 and TI3 lead to larger RMS values for the sunrise measurement in the morning where the simple approach performs better.





The total effect of using an intensity offset compensation is evaluated by TI4 which includes no offset correction. The resulting effect on $NO_2$ is almost the same as TI1 and TI3 (based on $I_0$) leading to the clear recommendation of using an offset compensation based on $I$ instead of $I_0$, which is supported by largely increased RMS values of TI4 (Fig. 10 bottom).

An intensity offset correction of first order (i.e. a term varying linearly with $\lambda$ in addition to a constant term) was tested in TI5, which is in practice often used not only for intensity offsets but also for compensation of the wavelength-dependence of the Ring slant column. The resulting $NO_2$ differences w.r.t. the reference fit (or TL0) are small with the exception of the first data point that is slightly off. In terms of RMS, TI5 leads to improvements of $\approx$20% in the morning. Interestingly, the TI5 RMS line shows some similarities to the IUPHD-IUPB line in Fig. 3d. However, no first order offset was included in the IUPHD fit, i.e. the offset implementation is not causing the observed similar shape in the morning (and the reason remains unclear). This is supported by increasing IUPHD RMS values in the evening in Fig. 3d, which are not present in TI5.

Fit TI6 includes again only a 0th order intensity offset, but a pseudo cross section accounting for the wavelength-dependence of the Ring slant column was added (the Ring cross section was multiplied by $\lambda$ and orthogonalized against the original cross section). The resulting RMS is indeed almost identical to TI5 (using a first order intensity offset) while the $NO_2$ is identical to TI0, i.e. the outlying first data point which is still partially present in TI5 disappeared in TI6. Thus, using a pseudo-cross section for compensation of the Ring wavelength-dependence seems preferable compared to using a first order intensity offset correction.

### 4.4 $I_0$-correction

In addition to the intensity offset tests, the effect of inclusion of an $I_0$-correction was evaluated in TI7 (which is identical to TI0 except for the $I_0$-correction, see Tab. 6). The respective line is shown in addition to the intensity offset investigations in Fig. 10. The $I_0$-effect adresses the problem that the limited instrument's resolution can cause an incomplete removal of Fraunhofer structures in the vicinity of strong narrow-banded absorption bands (Johnston, 1996; Wagner et al., 2001; Aliwell et al., 2002).

Only a very small constant offset in relative $NO_2$ slant column differences is obtained in TI7 ($\approx$0.25%, which is almost invisible in Fig. 10, middle). In terms of RMS, exclusion of the $I_0$-correction leads up to 20% increased RMS, which is comparable to different treatments of the intensity offset correction. Thus, inclusion of an $I_0$-correction is recommended in polluted environments such as the MAD-CAT site. It should be noted that the first data point is not an outlier in TI7, i.e. it is not sensitive to the $I_0$-correction.

### 4.5 Numerical methods (linear DOAS inversion)

The DOAS equation (Eq. 1) is a linear inverse problem

$$\mathbf{A}\,\mathbf{x} = \mathbf{b} \tag{5}$$

with the vector $\mathbf{x}$ (size $n$) containing the $n$ trace gas slant columns and polynomial coefficients of interest, the vector $\mathbf{b} = \ln(\frac{I_0}{I})$ (size m) containing the measured optical depths at $m$ wavelengths, and the $m \times n$ DOAS matrix $\mathbf{A}$ with columns consisting of absorption cross sections and polynomial terms $(1, \lambda, \lambda^2, \text{etc.})$ for the $m$ wavelengths.

Different numerical methods exist to solve Eq. 5 for $\mathbf{x}$. As $\mathbf{A}$ is non-square, no inverse exist. However, most retrieval codes calculate a pseudo-inverse $\mathbf{A}^{-1}$ (almost) fulfilling $\mathbf{A}^{-1}\mathbf{A} = \mathbf{I}$ (identity matrix) using a singular value decomposition (SVD) and obtain the slant columns of interest by $\mathbf{x} = \mathbf{A}^{-1}\,\mathbf{b}$. This method is frequently recommended for solving overdetermined linear inverse problems in terms of least squares (see e.g. Press, 1989).

However, after multiplying Eq. 5 with $\mathbf{A}^T$, the matrix $\mathbf{A}^T\mathbf{A}$ is quadratic and can be decomposed into an upper and a lower triangular matrix, $\mathbf{L}$ and $\mathbf{U}$. The linear inverse problem

$$\mathbf{A}^T\mathbf{A}\,\mathbf{x} = \mathbf{L}\mathbf{U}\,\mathbf{x} = \mathbf{A}^T\,\mathbf{b} \tag{6}$$



Table 7: Different methods tested for solving the linear DOAS equation $\mathbf{A\,x = b}$.

| Test | Retrieval code | Spectral grid | Method | Remarks |
|------|---------|---------|--------|---------|
| Reference | NLIN | $I_0$ | Pseudo-inverse of $\mathbf{A}$ using SVD | following Press (1989) |
| TL0 | Python | $I_0$ | Pseudo-inverse of $\mathbf{A}$ using SVD | different numpy and scipy implementations tested |
| TL1 | Python | $I_0$ | Solving quadratic $\mathbf{A^T A\,x = A^T\,b}$ using LU decomposition | different numpy and scipy implementations tested |
| TL2 | Python | $I_0$ | Invert $\mathbf{A^T A}$ using LU decomp. and multiply with $\mathbf{A^T\,b}$ | |
| TL3 | Python | 0.01 nm | same as TL0 | linear interpolation to 0.01 nm |
| TL4 | Python | 0.01 nm | same as TL0 | cubic spline interpolation to 0.01 nm |

can be solved then by forward substitution obtaining $\mathbf{y}$ from $\mathbf{L\,y = A^T\,b}$ and backward substitution obtaining $\mathbf{x}$ from $\mathbf{U\,x = y}$.

Furthermore, $\mathbf{A^T A}$ can also be directly inverted, normally by LU decomposition as well. The vector of slant columns is then obtained by $\mathbf{x = (A^T A)^{-1}\,A^T\,b}$. As the inversion takes normally much more computational steps, this method is known to be subject to roundoff errors, and therefore not recommended (Press, 1989).

The influence of these different numerical methods on resulting slant columns could not be easily
tested with the IUPB retrieval code and was thus evaluated in a Python script solving the DOAS equation using the same input (spectra, cross sections). Python (which is a well-established programming language in scientific computing) provides numerous different routines within its *numpy* and *scipy* packages (based on different subroutines from the *LAPACK* package, *http://www.netlib.org/lapack/*) that were tested for solving the DOAS equation. All performed tests are summarized in Tab. 7. Again,
differences of $NO_2$ slant columns and fit RMS have been calculated with respect to the IUPB v1 fit results (i.e. using NLIN). In order to restrict differences to the influence of numerical approaches only, the same slit function treatment as the IUPB retrieval code NLIN was applied in the Python script and the same (noon) reference spectrum was used. In addition, no further wavelength calibration, i.e. no nonlinear shift and squeeze fit was performed (neither in the Python script nor in the NLIN
reference fit used here) and no $I_0$-correction was included. The test results are shown in Fig. 11, again for the 2° elevation angle.

Results of test TL0 appear to be identical to the reference fit of the IUPB retrieval code, both using a SVD for inversion of the DOAS matrix. However, very small differences exist between TL0 and the reference fits (too small to be seen on the scale of Fig. 11), which are $< 0.006\%$ for $NO_2$ slant
columns and $< 0.07\%$ for RMS. These tiny disagreements can be attributed to numerical differences in programming languages.

TL1 and TL2 yield identical $NO_2$ slant columns (both are using an LU decomposition) which differ from SVD results by up to 0.7%. However, the RMS from TL2 was found to be an order of magnitude larger compared to the other tests and is therefore not shown in Fig. 11 (bottom). This is most likely
due to problems mentioned above, i.e. this finding is in agreement with common recommendations in textbooks. Interestingly, both TL1 (and TL2) $NO_2$ and RMS lines (which are differences w.r.t. IUPB) are similar in shape to the total fit RMS (compare to Fig. 3f). Thus, when the RMS increases, SVD inversion and LU decomposition lead to larger disagreements, both in RMS and $NO_2$. As the SVD yields smaller RMS values, it seems to be preferable, although the obtained improvement is only
about 2.5%.

Numerical differences may be obtained when performing the linear DOAS fit (Eq. 5) on another wavelength grid. Changes of the grid potentially arise from the wavelength calibration. Some retrieval codes (e.g. NIWA) also use an internal, equidistant wavelength grid. To test the effect of changes in





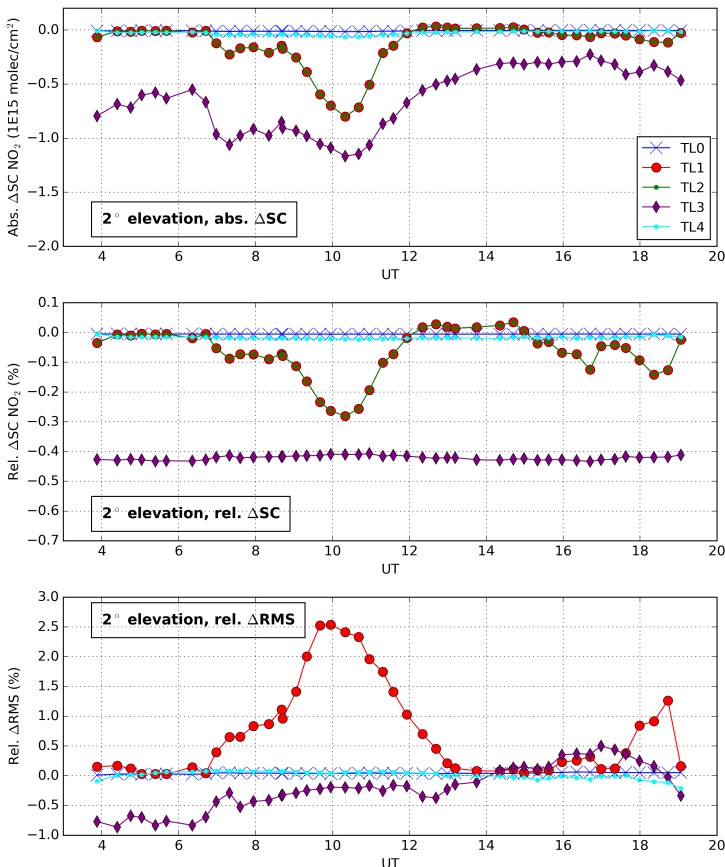

Figure 11: Absolute (top) and relative (middle) NO$_2$ slant column differences, and relative RMS differences (bottom) w.r.t. IUPB v1 results (only linear fit) in 2° elevation angle resulting from different numerical methods solving the linear DOAS equation.

the wavelength grid, the TL0 fit was repeated on an equidistant 0.01 nm grid (i.e. $I$, $I_0$, and cross sections were interpolated to 0.01 nm before solving Eq. 5). TL3 and TL4 are identical to TL0, but a linear interpolation was applied in TL3, while TL4 uses a cubic spline interpolation to 0.01 nm. Apparently, this results in a constant offset in relative NO$_2$ differences, which is seen most clearly in the TL3 line in Fig. 11 (middle). The resulting constant shift is $\approx 0.4\%$ for TL3, but only $\approx 0.02\%$ for TL4 meaning that the type of interpolation to the equidistant grid is of importance and the spline interpolation (not surprisingly) seems to resemble the spectrum better than a linear interpolation. However, using different wavelength grids might for example explain some of the observed differences between IUPB and NIWA, which were found to be constant in relative differences as well (see Fig. 3c). In terms of RMS, the computation on an equidistant 0.01 nm grid using linear interpolation behaves on average even a bit better (up to 1%). However, no recommendation can be drawn from this as discussed above.

## 4.6 Nonlinear wavelength calibration

As mentioned in Sect. 2.3, the spectra provided from the IUPB instrument were pre-calibrated using nightly HgCd line lamp measurements, which provide accuracies better than 0.1 nm. However, usually a post-calibration is included in DOAS retrieval codes in order to increase the fit quality (reducing



Table 8: Different wavelength calibration approaches evaluated.

| Test | Retrieval code code | Shift $I_0$ to Fraunhofer atlas | Shift $I$ to $I_0$ | Remarks |
|------|------|------|------|------|
| Reference | NLIN_D | yes | yes | Alternating scheme, v1 fit settings |
| TW0 | NLIN_D | no | no | Linear DOAS fit only |
| TW1 | NLIN_D | no | yes | |
| TW2 | NLIN_D | yes | no | |
| TW3 | Python | yes | yes | same a TW0 |
| TW4 | Python | yes | yes | same a TW3, but linear interpolation |
| TW5 | Python | yes | yes | same a TW3, all trace gases included in Fraunhofer $I_0$ fit |

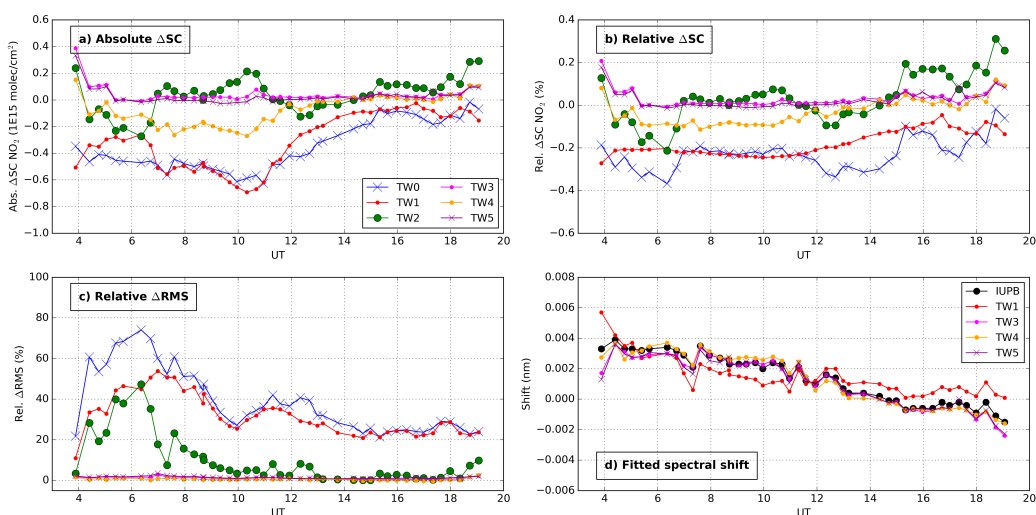

Figure 12: Absolute (a) and relative (b) $NO_2$ slant columns differences, and relative RMS differences (c) in 2° elevation angle resulting from different wavelength calibration approaches (Tab. 8). Corresponding shifts between $I$ and $I_0$ resulting from nonlinear fits are shown in (d).

RMS). This wavelength calibration is implemented in different ways in participating retrieval codes. Most groups calibrate the reference spectrum $I_0$ to a high resolution Fraunhofer atlas, apply the resulting calibration to all measured spectra, and allow in addition a shift and squeeze between $I$ and $I_0$ in order to compensate spectral shifts of the spectrometer during the day, e.g. caused by temperature changes. Note, KNMI uses a slightly different definition of the shift as discussed below.

This nonlinear shift and squeeze fit of the wavelength axis is mostly implemented in an iterative scheme together with the linear DOAS fit on $\ln(I_0/I)$. However, codes differ for example in whether trace gases are included in the shift and squeeze fit of $I_0$ to the high resolution Fraunhofer atlas. Sometimes also a higher order calibration is allowed or several sub-windows are used in order to characterize differently different parts of the spectra.

Tab. 8 summarizes tests performed to investigate the impact of different wavelength calibration approaches using the IUPB retrieval code NLIN. In addition, some tests were performed with the Python script form Sect. 4.5 which has therefore been extended to perform the nonlinear shift fit as not all tests could be easily implemented in the comprehensive NLIN software. Note, in contrast to the shift, the squeeze has been excluded from the intercomparison as it was found to be always 1.0 for

measurements shown here. Also QDOAS-specific implementations were not tested here.



Fig. 12 shows the resulting impact on $NO_2$ and RMS as well as the fitted shift between $I$ and $I_0$ (not present in all tests). $NO_2$ and RMS are again differences relative to the IUPB v1 fit results (without $I_0$-correction as this was not implemented in the Python routine). As in Fig. 3, the shift in Fig. 12d is no difference, and for comparison the shift from the reference fit is shown explicitly in black. The (fixed) shift retrieved from the nonlinear fit of $I_0$ to the Fraunhofer atlas is -0.035 nm. This is roughly a factor of 10 larger than the fitted shift between $I$ and $I_0$ shown in Fig. 12d. It is interesting to note that the shift is not zero around noon (time of the reference spectrum), indicating correlations between shift fit and other effects (predominantly intensity offset correction) and also indicating the presence of the tilt effect (Lampel et al., 2016).

The most extreme test is TW0 which excludes both the calibration of $I_0$ to the Fraunhofer atlas as well as the shift between $I$ and $I_0$. When omitting both calibration steps, the RMS is largely enhanced by up to 80% peaking in the morning and decreasing towards noon with a second smaller maximum around 12 UT. Interestingly, a very similar shape is seen in the RMS differences of INTA, KNMI, NUST, UNAM, and USTC in Fig. 3d. Although all of these groups are performing a wavelength calibration, TW0 indicates that differences in the calibration procedure are causing most of the disagreements between groups in terms of RMS. This is in contrast to $NO_2$ where changes of only 0.4% are obtained from TW0.

TW1 still excludes the absolute calibration to the Fraunhofer atlas, but includes the shift fit between $I$ and $I_0$. As seen before, the impact on $NO_2$ is very small ($\approx 0.4\%$), but absolute $NO_2$ differences of TW1 reflect the shape of total RMS and $NO_2$ slant columns (compare to Fig. 3), i.e. the relative differences are smooth in shape. The RMS is similar in shape as TW0, but the morning maximum is slightly later at 7 UT, the noon maximum around 11 UT and a small maximum in the evening occurs at 18 UT. This shape is similar to the relative RMS of NUST, UNAM, and NIWA in Fig. 3, but absolute numbers are different. However, this behaviour indicates that differences in the fitted Fraunhofer shift are partially responsible for observed differences between these and other groups. Interestingly, the fitted shift between $I$ and $I_0$ of TW1 in Fig. 12d is very similar to corresponding values of the reference fit, which is because the missing Fraunhofer shift is a different effect than the shift between $I$ and $I_0$.

In contrast to TW1, TW2 includes the Fraunhofer shift fit, but excludes the shift between $I$ and $I_0$. As this is the larger effect (-0.035 nm compared to only $\approx 0.004$ nm), the RMS is much smaller than in TW1 with a single maximum (up to 50%) in the early morning at 6 UT. The RMS timeseries shape is similar to the KNMI line in Fig. 3. The reason is a different definition of the shift in the KNMI retrieval: While in most retrievals $I$ is shifted relative to $I_0$, KNMI calculates the optical depth $\tau = \ln(I_0/I)$ without any shifts but allows then a shift of all cross sections relative to $\tau$. This is in first order compensating the effect of the Fraunhofer shift but neglecting potential shifts between $I$ and $I_0$ (in this case Fraunhofer lines would not cancel out completely in the optical depth $\tau$). As a result, the KNMI approach is similar (but not identical) to TW2. The fit quality following the KNMI approach is expected to be better using a sequential reference as the temperature drift of the spectrometer is much smaller then. This matches perfectly with observations in Figs. 4 and 6 showing a better agreement between KNMI and IUPB when using a sequential reference. However, it has to be mentioned that the change of $NO_2$ in TW2 is small (0.3%) compared to the change in fit RMS (50%).

TW3 uses the same wavelength calibration treatment as the IUPB reference fit, but is performed in another programming code (Python) evaluating how much difference is caused by use of another programming code and numerical issues. The fitted shift is mostly identical to the reference fit except for the early morning and late evening when also the $NO_2$ shows very slight differences. The largest disagreement of $NO_2$ is 0.2% for the first measurement of the day that was affected by large stray light effects (and potentially direct light) and thus most likely indicating cross-correlations between shift fit and intensity offset correction. The resulting RMS is almost identical to the reference fit (Fig. 12c).

TW4 is the same as TW3 but the spline interpolation (calculating $I$ at spectral points of $I_0$ during the nonlinear shift fit) is replaced by a simple linear interpolation. The fitted shift is changed slightly and $NO_2$ differences are up to 0.1% which is of the same order than when not performing any shift fit at all demonstrating that the shift fit has a negligible impact on $NO_2$. In contrast, it has a large impact on RMS, where the marginally different methods between TW3 and TW4 produce the same



Table 9: Summary of performed tests (differences in retrieval codes) and associated impacts on $NO_2$ slant columns and RMS.

| Reason for disagreement | $\Delta NO_2$ (%) | $\Delta RMS$ (%) | Remarks |
|---|---|---|---|
| Reference treatment (noon) | 2.5 | 3 | Produces constant absolute $NO_2$ SC offsets. |
| Reference treatment (seq.) | 8 | 6 | |
| Slit function treatment | 1.3 | 6.5 | Produces constant relative $NO_2$ SC offsets. |
| Intensity offset correction | 2 (typically) 10 (outlier) | 20 (typically) 60 (outlier) | |
| $I_0$ correction | 0.25 | 20 | Produces constant relative $NO_2$ SC offsets. |
| Numerical methods (for linear DOAS fit) | 0.3 (0.7) | 2.5 | Produces constant relative $NO_2$ SC offsets. Disagreements increase with RMS. |
| Wavelength calibration (nonlinear shift fit) | 0.4 | up to 80 | |

RMS while the effect of excluding the shift completely leads to largely enhanced RMS.

In the Fraunhofer calibration of the reference fit using NLIN (nonlinear shift fit of $I_0$) as well as in TW3 all trace gas absorptions are omitted, i.e. an iterative scheme between shift fit and DOAS fit comprising only a polynomial of order 4 is applied. In contrast, all trace gas absorptions are included in the Fraunhofer calibration in TW5. As a result, the RMS of the DOAS fit between Fraunhofer spectrum and $I_0$ is reduced by a factor of 2 and the nonlinearly fitted shift is -0.031 nm instead of

-0.035 nm in the reference fit. However, this has only a marginal influence on $NO_2$, RMS and fitted shift between $I$ and $I_0$ in Fig. 12 and consequently explains none of the observed differences between groups.

It is important that all shifts (even TW1) in Fig. 12d show the same general shape that is also retrieved by most groups in Fig. 3. Only shifts of INTA and KNMI could not be reproduced by any

of the performed tests (but because of the different definition, the KNMI shift is not expected to match with other groups results). However, RMS shapes in Fig. 12c suggest that differences in the wavelength calibration are the major reason of observed RMS disagreements between groups in Fig. 3.

## 5 Summary and conclusion

An intercomparison of DOAS retrieval codes using measured spectra from the same instrument during

the MAD-CAT campaign and harmonized fit settings was performed. Excellent agreement was found between different DOAS fit algorithms from 17 international groups. For noon reference fits, the correlation in terms of $NO_2$ slant columns was found to be larger ($> 99.98\%$) than for sequential references ($> 99.2\%$), which is caused by different implementations of the sequential reference. For individual measurements, differences of up to 8% in resulting $NO_2$ slant columns (which is substantially

larger than $NO_2$ slant column fit errors), and up to 100% for the fit RMS were observed.

Interestingly, groups using the same retrieval code do not always produce results showing the same systematic behavior - except QDOAS users when performing sequential reference fits (which is because the selection of the sequential reference was implemented in a fixed way in those QDOAS versions). This is the result of different options - other than the harmonization settings agreed on - users select.

A survey of participating DOAS retrieval codes revealed five potential reasons causing differences in fit results, which were investigated in more detail. Typical impacts on $NO_2$ slant columns and RMS arising from these different implementations/options in DOAS retrieval codes are summarized in Tab. 9.

In general, the wavelength calibration and the intensity offset correction were found to produce the

majority of observed RMS differences, but have a negligible impact on $NO_2$ slant columns ($< 0.4\%$, resp. $< 2\%$ except for the first measurement of the day affected by stray light and possibly direct light in the telescope). In contrast, the reference selection explains the majority of observed $NO_2$ slant column differences between groups while having a minor impact on the RMS. Thus, if harmonization of $NO_2$ slant columns is of interest, the reference treatment needs to be harmonized (otherwise differ-



ences of up to 8% have to be expected) while for RMS reduction/harmonization, the offset intensity correction and the wavelength calibration need to be harmonized.

In terms of $NO_2$, two types of disagreements between groups have been observed, which are (1) constant in absolute, or (2) constant in relative differences. The latter was found to arise from the numerical approach used for solving the DOAS equation as well as the treatment of the slit function 795 while the choice of the reference spectrum causes absolute differences.

Recommendations aiming at improvement of the fit quality and harmonization between MAX-DOAS retrievals derived from this study are:

1) Reference treatment: Using averaged or interpolated sequential reference spectra matches the atmospheric conditions at the measurement time better and was found to produce slightly smaller 800 RMS (6%).

2) Slit function: Using a measured slit function performed better than fitting line parameters in the data set used here. However, slit function measurements then have to be performed regularly (e.g., daily to monitor possible instrument changes).

3) Intensity offset: An approach based on $I$ instead of $I_0$ is recommended. Surprisingly, the simple 805 approach performs better for measurements pointing close to sunrise but this could be just a coincidence in this data set. Inclusion of an additional Ring spectrum multiplied by wavelength is preferred over adding a linear term to the offset as this mostly compensates the wavelength-dependence of the Ring slant column.

4) Numerical approaches: Using an SVD is most stable and produces slightly smaller RMS than 810 LU decomposition.

5) Wavelength calibration: Although HgCd line lamp calibration measurements lead to absolute accuracies (in this case) of $\approx 0.03$ nm, a Fraunhofer shift fit reduces the RMS by up to 40-50%. For the $I_0$ fit w.r.t. the Fraunhofer spectrum, inclusion of all trace gases showed no advantage over inclusion of a polynomial only. Temperature instabilities of the spectrometer produced shifts 815 of $I$ relative to $I_0$ changing over the day. Compensation of this effect within DOAS retrieval codes further improves the RMS by up to 40-50% when using a noon reference spectrum.

## 6 Acknowledgments

We thank the Max-Planck Institute for Chemistry, Mainz, for hosting and organizing the MAD-CAT campaign. This study has been performed in the context and with financial support of the EU-820 QA4ECV project. CU Boulder furthermore acknowledges B. K. Dix for support. The work of the IAP group was supported by the Russian Science Foundation by grant 14-47-00049.

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
