# Peer review of "Investigating differences in DOAS retrieval codes using MAD-CAT campaign data"

_Atmospheric Measurement Techniques, 2016_

## Referee Comment (RC1) · Anonymous Referee #1 · 27 Dec 2016

Review of **Investigating differences in DOAS retrieval codes using MAD-CAT campaign data**, by E. Peters *et al.*

AMTD, doi:10.5194/amt-2016-358, 2016

**General Evaluation**

Peters et al. present a comparison study on NO2 slant column retrievals from a range of different DOAS retrieval codes on the same set of MAX-DOAS observations acquired during the 2013 MAD-CAT campaign. Results of NO2 columns and RMS values from the retrieval codes, which are run with a basic set of harmonized settings for molecular absorption cross-sections and closure polynomials, are compared and a range of possible sources for their differences is investigated. Based on this study, a short list of recommendations is given as general guidance for DOAS retrievals to obtain high confidence/low RMS retrievals.

The paper is solidly written, and there is little to criticize in methodology and overall quality of presentation. My main criticism is that, at 27 journal pages, the manuscript is overly long for a study that concludes with five basic recommendations. The paper provides excellent insight into the workings of DOAS retrievals, and as such is valuable for both data providers and data users, but this reviewer strongly suggests that the discussion be tightened and the main part of the message conveyed more concisely.

The meat of the paper is straight forward and relatively simple:
1.  A set of MAX-DOAS observation from MAD-CAT was selected for a retrieval algorithm comparison.
2.  A common set of basic spectral fitting settings was prescribed with which to run the retrieval codes.
3.  Differences in NO2 slant columns and RMS were found, relative to a reference retrieval.
4.  The reference retrieval code was run with modified settings for five essential code elements - radiance reference spectrum, slit function, offset correction, I0 correction, matrix inversion - to investigate their effect on the retrievals and to possibly explain the differences in the results.
5.  The modified retrieval runs lead to the final recommendations, while the attempt to attribute differences between the codes to the investigated five sources is only moderately successful.

The majority of the "take home messages" comes from Bullet 4 above, but the attribution of those effects in the actual differences observed between the results from the various retrieval codes remains qualitative at best. With this in mind, any figures and discussions relating primarily to relationships between the results from different codes - in particular figures 4 and 6 and their discussion - are non-essential and should be marked for removal.

Below are some more specific comments. Very few of these are copy-editorial, since the level of presentation of this paper is very high.

**Recommendation**

The manuscript is acceptable for publication, but should undergo some tightening and add a few clarifications. Since there are no basic problems with methodology or presentation, a second round of review is not necessary.

**Specific Comments**

*Retrieval Uncertainties*

While the paper compares NO2 slant columns and retrieval RMS, no NO2 slant column uncertainties are shown. Purely spectral minimization-based uncertainties are a combination of RMS and fitting covariances, and thus provide important information on the quality of the retrieved slant columns beyond the RMS.

*Reference Cross-Section Wavelength Scale*
Three (admittedly very basic) questions regarding wavelength registration:
1. Does the IUPB MAX-DOAS instrument measure in vacuum or air?
2. Which wavelength registration (vacuum or air) was used for the retrievals?
3. Was it assured that all molecular and solar reference spectra were on the same type of wavelength registration as the MAX-DOAS spectra?

*Slit Function*
The measured slit function as shown in Figure 8 is slightly asymmetric. Yet, no attempts are reported of having fit an asymmetric Gaussian to the measurement for use in the retrievals. At least part of the comparison exercise utilized pre-convolved molecular absorption cross-sections, so this should have been an easy case to include.  It is not very surprising that results from original and re-centered original slit function are virtually identical: the asymmetry should mainly manifest as a spectral shift, which is taken care of by the shift parameter during the retrieval process.
Regarding the differences introduced by removing the offset of 0.001: was the resulting slit function renormalized to the same area as the one with the offset?

*Section 4 "Understanding differences between retrieval codes"*
Ultimately, this is the most important section of the manuscript since it systematically investigates the effect of different fit settings on the retrieved slant columns and the resulting RMS. It is also here that the five recommendations in the Summary are derived. In principle, this exercise is independent of the MAD-CAT comparison. While the differences in results from the various retrieval codes are a good motivation to perform these tests, they are valuable in their own right, and more emphasis should be placed on this. By referring to this part of the study as "differences between retrieval codes", this reviewer believes that the importance of these tests is somewhat muddled and degraded. The reader would benefit from a clear statement of the type "differences between the harmonized MAD-CAT retrieval results prompted the team to systematically investigate effects of the non-harmonized aspects of the retrievals, with the aim to derive a key set of Best Practices recommendations". Since quantitative attribution of "what part of the differences originates from which non-harmonized retrieval setting" turns out to be unfeasible/unsuccessful, more emphasis should be placed on the derived recommendations for DOAS retrievals.

**Editorial Comments**
Line 122: suggest to reword as "real data without cross-instrumental bias", to avoid confusion with measurements free of instrumental bias.

Figure 1 caption: suggest to include "(90° = Zenith)" for the benefit of readers less familiar with MAX-DOAS observation methodology.

Figure 2: Use a different color for the fitted spectrum. Green and Blue are hard to distinguish.

Figure 2: What is the definition of "differential cross section", and is it optical "density", "depth", or "thickness"? None of these quantities would be expected have negative values, thus there has to be a reference point.

Line 138: "However, these are normally the ones of interest".

Line 171: "$r$, the root mean square (RMS) of the fit residual, is an important quantity used within this study to identify and evaluate differences between the DOAS retrieval codes."

Line 229: either "groups participating in MAD-CAT" or "participating MAD-CAT groups".

Line 305: delete "one" after "WCRS".

---

## Referee Comment (RC2) · Anonymous Referee #2 · 10 Jan 2017

General Evaluation:

The paper by Peters et al. investigates differences in NO2 DOAS retrieval due to differences in different DOAS retrieval codes. A same set of MAX-DOAS observations and DOAS retrieval settings were provided to various groups using different DOAS retrieval codes to retrieve NO2 differential slant column densities (dSCD). Resulting NO2 dSCD and RMS were then compared and a range of possible sources for their differences were investigated. In the end, the authors provide a list of 5 recommendations aimed at improving the NO2 DOAS retrievals. These recommendations are fairly straight forward and largely constitute best practices that are already followed to improve DOAS retrievals. Overall, the paper is very nicely written but way too long. I suggest the authors make changes to shorten the length of the paper.

[Figure]

Recommendation:

The paper fits the scope of AMT journal as it documents differences is DOAS retrieval codes and hence the paper is acceptable for publication in AMT with modifications to (i) make the differences between the codes more explicit and (ii) shorten the manuscript.

Specific Comments:

There are two parts to the paper and they could very easily be two separate papers. The first part focuses on the intercomparison of NO2 from different retrieval codes. While in the surface it seems like a nice and interesting idea to compare different DOAS retrieval codes, it becomes very clear after section 3.1 (Part 1) that differences due to retrieval codes are very small compared to differences due to instrumental design (see Roscoe et al, 2010). The second part investigates the potential sources of these differences. However it appears like a sensitivity study to determine best features to have in a retrieval code. The authors performed the sensitivity tests first, and then compared the results from the sensitivity tests to the results from various groups. Based on the results the authors went back to the groups to verify their findings. There is disconnect between section 3 and 4 as there is lack of basis for the tests being performed. I suggest the authors include an overview table (using the survey data) which highlights the differences between different retrieval codes to connect section 3 and 4. This table could replace most of the text describing the retrieval codes.

The amount of details for different retrieval codes are not comparable. Some codes are described in details while others (e.g. IUPHD) barely include a sentence.

The authors make 5 recommendations to improve fit quality and harmonization between MAX-DOAS retrievals. Is there one/many DOAS retrieval code which already have these features? If so please include this/these codes as the current state of the art. This would be especially useful for new users.

Why does the reference after the scan (T6 settings) results in larger differences? Is it

simply due to the time difference between 2 degree EA and reference spectra? What is the time difference between the two spectra? Also do you see similar behavior between references taken before the scan and spectra further away? For example between refA and spectra EAnA, or refA and spectra EA2B in the following scan sequence (refA, EA2A, . . . , EAnA, refB, EA2B,. . . ., EAnB, refC, . . ..).

There is no specific need to include all the QDOAS results in the paper. I suggest the authors consolidate the QDOAS results. This could be done by either presenting select QDOAS results or grouping all QDOAS results together for clarity (e.g. similar symbol in figure 3 or one side of the plot in figure 4). It would help compare and contrast the results between QDOAS and other codes.

Line 790: "differences of up to 8% have to be expected" – Does this also hold true for other elevation angles where dSCDs are smaller? To some extent quoting 8% as expected uncertainty is somewhat misleading knowing that the particular spectrum was affected by direct sunlight and such a scenario is not common in MAX-DOAS measurements. I suggest the authors make this distinction clear in the manuscript in order to avoid misuse of 8% as inherent uncertainty in DOAS retrievals.

---

## Author Comment (AC1) · 31 Jan 2017

*Point-by-point answers to Anonymous Reviewer #1*

*Note: Reviewers comments are printed in black, author's replies in blue (and italic).*

**General Evaluation**

Peters et al. present a comparison study on NO2 slant column retrievals from a range of different DOAS retrieval codes on the same set of MAX-DOAS observations acquired during the 2013 MAD-CAT campaign. Results of NO2 columns and RMS values from the retrieval codes, which are run with a basic set of harmonized settings for molecular absorption cross-sections and closure polynomials, are compared and a range of possible sources for their differences is investigated. Based on this study, a short list of recommendations is given as general guidance for DOAS retrievals to obtain high confidence/low RMS retrievals.

The paper is solidly written, and there is little to criticize in methodology and overall quality of presentation. My main criticism is that, at 27 journal pages, the manuscript is overly long for a study that concludes with five basic recommendations. The paper provides excellent insight into the workings of DOAS retrievals, and as such is valuable for both data providers and data users, but this reviewer strongly suggests that the discussion be tightened and the main part of the message conveyed more concisely.

*We would like to thank the reviewer for these general encouraging comments and evaluation. We also fully agree with the main criticism concerning the length of the study and tried to shorten it following the suggestions provided by the reviewer (see below).*

The meat of the paper is straight forward and relatively simple:

1. A set of MAX-DOAS observation from MAD-CAT was selected for a retrieval algorithm comparison.
2. A common set of basic spectral fitting settings was prescribed with which to run the retrieval codes.
3. Differences in NO2 slant columns and RMS were found, relative to a reference retrieval.
4. The reference retrieval code was run with modified settings for five essential code elements - radiance reference spectrum, slit function, offset correction, I0 correction, matrix inversion – to investigate their effect on the retrievals and to possibly explain the differences in the results.
5. The modified retrieval runs lead to the final recommendations, while the attempt to attribute differences between the codes to the investigated five sources is only moderately successful.

   *Please note, INTA and NIWA found a fault in their retrieval code (which is a benefit of the present study) and repeated their data analysis, after the paper was published in AMT discussions. The new results are included in the revised manuscript as INTA2 and NIWA2 and a lot of differences, which could not be attributed to any of the performed tests before, disappeared. With this, almost all observed differences between groups could now be attributed to the performed tests (or attributed to faults in retrieval codes which are now corrected).*

The majority of the "take home messages" comes from Bullet 4 above, but the attribution of those effects in the actual differences observed between the results from the various retrieval codes remains qualitative at best. With this in mind, any figures and discussions relating primarily to relationships between the results from different codes - in particular figures 4 and 6 and their discussion - are non-essential and should be marked for removal.

*The attribution is much more successful in the revised manuscript after faults in retrieval codes have been corrected (see above), which is good for the "detection rate" but also good for participating groups. We agree that the paper needs to be shortened, but in our opinion the statistics reflected in figure 4 (and 6)*

*are essential as they document quantitatively the agreement between retrieval codes that can be expected – without them the manuscript would be purely qualitative. Therefore, we decided to skip figure 6 (with corresponding text) to tighten the manuscript, but to leave figure 4 demonstrating statistics and the quantitative agreement between groups. The basic results of figure 6 as well as results of different fit settings (which were not included as figures in the manuscript at all) are then summarized in Table 3 (which needs at least one figure to be understandable for the reader).*

Below are some more specific comments. Very few of these are copy-editorial, since the level of presentation of this paper is very high.

**Recommendation**
The manuscript is acceptable for publication, but should undergo some tightening and add a few clarifications. Since there are no basic problems with methodology or presentation, a second round of review is not necessary.

**Specific Comments**

**Retrieval Uncertainties**
While the paper compares NO2 slant columns and retrieval RMS, no NO2 slant column uncertainties are shown. Purely spectral minimization-based uncertainties are a combination of RMS and fitting covariances, and thus provide important information on the quality of the retrieved slant columns beyond the RMS.
*This is in general true. However, slant column errors are calculated from RMS, SCs, and covariance of the (pseudo-inverted) DOAS matrix. For this exercise, the DOAS matrix is (ideally) the same for all groups, as fit settings were prescribed and the same cross sections were used distributed to all groups (see above). Thus, an investigation of the fit errors would provide very little new insight beyond the comparison performed for SCs and RMS.*
*Nevertheless, an important aspect is how large slant column disagreements between groups are in comparison to slant column errors. This is mentioned in Sect 3.1 (stating in summary that $NO_2$ slant column differences between groups were found to be up to 2-3 times larger than typical slant column errors). We agree that this is a remarkable finding, and put it also in the conclusions of the revised manuscript.*

**Reference Cross-Section Wavelength Scale**
Three (admittedly very basic) questions regarding wavelength registration:
1. Does the IUPB MAX-DOAS instrument measure in vacuum or air?
*It measures in air.*
2. Which wavelength registration (vacuum or air) was used for the retrievals?
*Wavelength in air.*
3. Was it assured that all molecular and solar reference spectra were on the same type of wavelength registration as the MAX-DOAS spectra?
*All cross sections were converted to wavelength in air (unless they were already given in wavelength in air) before they were distributed to participating groups. It was prescribed to use the provided cross sections as summarized in Tab. 1. (However, even a potential error in the wavelength-conversion would affect all groups in the same way and should therefore have no influence on the objective of this study).*

**Slit Function**
The measured slit function as shown in Figure 8 is slightly asymmetric. Yet, no attempts are reported of

having fit an asymmetric Gaussian to the measurement for use in the retrievals. At least part of the comparison exercise utilized pre-convolved molecular absorption cross-sections, so this should have been an easy case to include. It is not very surprising that results from original and re-centered original slit function are virtually identical: the asymmetry should mainly manifest as a spectral shift, which is taken care of by the shift parameter during the retrieval process.

Regarding the differences introduced by removing the offset of 0.001: was the resulting slit function renormalized to the same area as the one with the offset?

*Yes, the slit function was of course re-normalized after an offset was subtracted. However, there seems to be a small misunderstanding as the value of 0.001 is not the subtracted offset (first column of Table 5) but the cut-off value (tests summarized in third column of Table 5) for the discrete convolution.*

*The reviewer is right, different centered slit functions should result in a shift of cross sections. However, in the DOAS test fits performed in this section, no shift between cross sections and optical depth was allowed (but the shift between I and I0 will partly compensate for it). We included this statement and generally rephrased the whole section in order to increase readability and clarity.*

*Asymmetric slit function test: It is correct that some retrieval codes offer the possibility to fit more sophisticated line shape parameters taking into account potential asymmetry (as seen here) much better. The detailed use of the slit function was not included in the prescribed fit settings as being implemented differently in different retrieval codes. The tests performed here demonstrate in summary two extreme scenarios which characterize the maximum difference resulting from the slit function treatment, namely (1) using the (asymmetric) slit function as it is, or (2) fitting basic line parameters. The use of an asymmetric fitted slit function would clearly lead to a result in between these scenarios, but would not help a lot for the attribution of differences between groups to their sources as the exact implementations could not be reproduced here. We included this explicitly in the revised manuscript.*

*Section 4 "Understanding differences between retrieval codes"*

Ultimately, this is the most important section of the manuscript since it systematically investigates the effect of different fit settings on the retrieved slant columns and the resulting RMS. It is also here that the five recommendations in the Summary are derived. In principle, this exercise is independent of the MAD-CAT comparison. While the differences in results from the various retrieval codes are a good motivation to perform these tests, they are valuable in their own right, and more emphasis should be placed on this. By referring to this part of the study as "differences between retrieval codes", this reviewer believes that the importance of these tests is somewhat muddled and degraded. The reader would benefit from a clear statement of the type "differences between the harmonized MAD-CAT retrieval results prompted the team to systematically investigate effects of the non-harmonized aspects of the retrievals, with the aim to derive a key set of Best Practices recommendations". Since quantitative attribution of "what part of the differences originates from which non-harmonized retrieval setting" turns out to be unfeasible/unsuccessful, more emphasis should be placed on the derived recommendations for DOAS retrievals.

*We thank the reviewer for this good suggestion and changed section 4, now called "Sensitivity studies of non-harmonized retrieval aspects" accordingly. Similar changes have been made in the conclusions, the introduction, and the abstract as well. However, we think the MAD-CAT intercomparison results are more than a good motivation leading to these tests because (as mentioned above) after finding and eliminating additional faults in retrieval codes (unfortunately after publication in AMT discussions), almost all systematical differences between groups could now be attributed to sources found in section 4.*

**Editorial Comments**

Line 122: suggest to reword as "real data without cross-instrumental bias", to avoid confusion with measurements free of instrumental bias.
*We rephrased this as: "The work reported here overcomes limitations from previous studies by using real measurements originating from a single instrument. This facilitates the study of the agreement between different retrieval codes on real data without instrumental biases between results from different groups."*

Figure 1 caption: suggest to include "(90° = Zenith)" for the benefit of readers less familiar with MAX-DOAS observation methodology.
*Thanks, we changed that figure accordingly.*

Figure 2: Use a different color for the fitted spectrum. Green and Blue are hard to distinguish.
*We changed the blue dashed line to red and increased the linewidth.*

Figure 2: What is the definition of "differential cross section", and is it optical "density", "depth", or "thickness"? None of these quantities would be expected have negative values, thus there has to be a reference point.
*The differential cross section is the absolute cross section minus a (fitted) polynomial. We clarified this in the revised manuscript. The optical density shown here is the differential optical density (same definition as above), which is also explicitly mentioned in the revised manuscript. In general, optical "thickness", "depth" and "density" are often mixed-up in the literature. Within this manuscript, we tried to use only "optical depth" (i.e. avoid thickness) and used "optical density" when referring to one specific trace gas only (while "depth" is used for the total absorption effects of all trace gases, i.e. the measured quantity ln(I/I0)).*

Line 138: "However, these are normally the ones of interest".
*Changed.*

Line 171: "*r*, the root mean square (RMS) of the fit residual, is an important quantity used within this study to identify and evaluate differences between the DOAS retrieval codes."
*No, r is not the root mean square of the fit residual, but it is the fit residual itself (as denoted in the corresponding equation and explained in the text). So the text was correct (remains unchanged).*

Line 229: either "groups participating in MAD-CAT" or "participating MAD-CAT groups".
*Thanks, changed to "groups participating in MAD-CAT".*

Line 305: delete "one" after "WCRS".
*Thanks, we deleted "one".*

---

## Author Comment (AC2) · 31 Jan 2017

*Note: Reviewers comments are printed in black, author's replies in blue (and italic).*

General Evaluation:
The paper by Peters et al. investigates differences in NO2 DOAS retrieval due to differences in different DOAS retrieval codes. A same set of MAX-DOAS observations and DOAS retrieval settings were provided to various groups using different DOAS retrieval codes to retrieve NO2 differential slant column densities (dSCD). Resulting NO2 dSCD and RMS were then compared and a range of possible sources for their differences were investigated. In the end, the authors provide a list of 5 recommendations aimed at improving the NO2 DOAS retrievals. These recommendations are fairly straight forward and largely constitute best practices that are already followed to improve DOAS retrievals. Overall, the paper is very nicely written but way too long. I suggest the authors make changes to shorten the length of the paper.

Recommendation:
The paper fits the scope of AMT journal as it documents differences is DOAS retrieval codes and hence the paper is acceptable for publication in AMT with modifications to (i) make the differences between the codes more explicit and (ii) shorten the manuscript.
*We like to thank the reviewer for the generally encouraging comments. We also share the opinion that the manuscript should be shortened. We did this following the specific reviewer's suggestions as described below.*

Specific Comments:
There are two parts to the paper and they could very easily be two separate papers. The first part focuses on the intercomparison of NO2 from different retrieval codes. While in the surface it seems like a nice and interesting idea to compare different DOAS retrieval codes, it becomes very clear after section 3.1 (Part 1) that differences due to retrieval codes are very small compared to differences due to instrumental design (see Roscoe et al, 2010). The second part investigates the potential sources of these differences. However it appears like a sensitivity study to determine best features to have in a retrieval code. The authors performed the sensitivity tests first, and then compared the results from the sensitivity tests to the results from various groups. Based on the results the authors went back to the groups to verify their findings. There is disconnect between section 3 and 4 as there is lack of basis for the tests being performed. I suggest the authors include an overview table (using the survey data) which highlights the differences between different retrieval codes to connect section 3 and 4. This table could replace most of the text describing the retrieval codes.
*We changed the motivation for Part2 to make a stronger connection and better explanation for Part 2 following from Part1 (we did this already in response to comments of Anonymous Reviewer #1).*
*It is not true that we performed the sensitivity tests first. The intercomparison between groups was performed first, then a survey to identify possible reasons for observed differences, and then the sensitivity studies in order to evaluate the effect of each of the potential reasons. We state this more clearly in the revised manuscript.*
*It is true that differences are smaller than observed in Roscoe et al. 2010 and we agree that instrumental differences are supposed to be larger. However, we cannot strictly conclude that remaining differences come from instrumental design alone. For example, during CINDI (Roscoe et al, 2010), the instruments did not point at exactly the same time into the same direction (and also in real measurements all kinds of misalignments are potentially present). In this aspect, the recent CINDI-2 campaign is interesting because instruments followed a strict measurement protocol, i.e. coinciding measurements are assured. However, there are no CINDI-2 results published yet.*

*In order to meet the reviewer's comments, we changed and shortened Sect. 2.4. In particular, instead of the list of participating groups we now introduce each retrieval code and emphasize important differences (we also tried to put all content into a table, as suggested by the reviewer, but it turned out that the table format is not suited). We did not include an additional table for the survey as this would again increase the length of the manuscript while providing little new insight. We therefore directly mention the result of the survey leading to the 5 systematic differences which are then subject of the performed sensitivity tests.*

The amount of details for different retrieval codes are not comparable. Some codes are described in details while others (e.g. IUPHD) barely include a sentence.
*This whole section has been changed in the revised manuscript (see above). Explanations of retrieval codes still differ a bit in length, but sufficient references are always included.*

The authors make 5 recommendations to improve fit quality and harmonization between MAX-DOAS retrievals. Is there one/many DOAS retrieval code which already have these features? If so please include this/these codes as the current state of the art. This would be especially useful for new users.
*Of course some of the retrieval codes allow all recommended options, e.g. NLIN (obviously, as it was used for performing the sensitivity tests) or QDOAS in its latest version (which features the use of an interpolated reference spectrum). However, it is a bit delicate to nominate one or a subset of participating retrieval codes as the current state of the art. It is also not the objective of this work to recommend any retrieval for new users (also, some of them are in-house software and not even publicly available). Moreover, while some of the recommended best practices are intrinsic features of the code (e.g. numerical computation) others are normally options for the user (e.g. choice of background spectrum). The given recommendations are therefore valuable for both, users (best practices independent from specific code) as well as designers (intrinsic features) that either programmed the participating codes (some faults have already been found and corrected within this study) or intend to design an own retrieval in the future.*

Why does the reference after the scan (T6 settings) results in larger differences? Is it simply due to the time difference between 2 degree EA and reference spectra? What is the time difference between the two spectra? Also do you see similar behavior between references taken before the scan and spectra further away? For example between refA and spectra EAnA, or refA and spectra EA2B in the following scan sequence (refA,EA2A, . . . , EAnA, refB, EA2B,. . .., EAnB, refC, . . ..).
*Please notice that this question affects fit TR6 in Fig. 7, which is in the revised manuscript Fig. 6 as one of the previous plots have been removed in order to shorten the manuscript (following a suggestion from Reviewer #1).*
*In the reference fit used for differences plotted here the $I_0$ spectrum is the zenith spectrum before and after the scan interpolated to the measurement time. As the vertical scanning sequence starts from low to high elevations, the one before is closer to the measurement time (additional comment: tests TR4 and TR5 are therefore almost identical) than the one after the scan. So the reviewer is correct. Unfortunately, all options for the sequential reference that are currently implemented in our retrieval code are shown in Fig. 6 and at the moment no option is possible to test the referee's question. Nevertheless, it is very likely that references further away but before the scan would result again in larger differences.*

There is no specific need to include all the QDOAS results in the paper. I suggest the authors consolidate the QDOAS results. This could be done by either presenting select QDOAS results or grouping all QDOAS results together for clarity (e.g. similar symbol in figure 3 or one side of the plot in figure 4). It would help compare and contrast the results between QDOAS and other codes.

*We partly agree and followed the referee's suggestion indicating all QDOAS groups by the same symbol in Fig. 3 and 5 (formally Fig. 6) for clarity and better comparison to non-QDOAS groups. We also indicated QDOAS groups in Fig. 4 for the same reason.*

*However, we do not think that consolidating all QDOAS results is a good option because groups using QDOAS show clearly different results and no systematic similar pattern, which is an interesting finding that is explicitly mentioned in the conclusion section and has some impact for other studies: The amount of prescribed fit settings in this study is comparable to intercomparison campaigns like CINDI or CINDI-2. Consequently, groups participating those campaigns will provide intrinsic (and non-systematic) differences in their results due to small differences in non-harmonized (detailed) settings (even if using the same retrieval code), which have to be expected in the same range as observed here. We point this out more clearly in the revised manuscript. Finally, the QDOAS versions used here are different.*

Line 790: "differences of up to 8% have to be expected" – Does this also hold true for other elevation angles where dSCDs are smaller? To some extent quoting 8% as expected uncertainty is somewhat misleading knowing that the particular spectrum was affected by direct sunlight and such a scenario is not common in MAX-DOAS measurements. I suggest the authors make this distinction clear in the manuscript in order to avoid misuse of 8% as inherent uncertainty in DOAS retrievals.

*The value of 8% does not correspond to the first data point in Fig. 3 (although this holds true as well) but to the sequential references seen in Fig. 5 where disagreements of 8% are frequently obtained. We stated this more clearly in the revised manuscript.*

*In general, the value of 8% is representative for small elevations as Figure 1 (below) shows. Fig. 5 (formally Fig. 6) in the paper has been reproduced here, but this time for 8° elevation instead of 2°. Although absolute differences between groups appear to be a bit smaller in 8° than in 2°, relative differences are even a bit larger because slant columns are smaller (but nevertheless, 8% appears to be reasonable even for 8° elevation measurements shown here).*

[Figure]

*Figure 1: NO$_2$ differences between groups for 8° elevation angle.*